# The decoration of specialized metabolites influences stylar development

**Jiancai Li[1], Meredith C Schuman[1,2], Rayko Halitschke[1], Xiang Li[1], Han Guo[1], Veit Grabe[3], Austin Hammer[4], Ian T Baldwin[1]\***

[1]Department of Molecular Ecology, Max Planck Institute for Chemical Ecology, Jena, Germany; [2]Department of Geography, University of Zurich, Zurich, Switzerland; [3]Department of Evolutionary Neuroethology, Max Planck Institute for Chemical Ecology, Jena, Germany; [4]Department of Biology, Brigham Young University, Provo, United States

**Abstract** Plants produce many different specialized (secondary) metabolites that function in solving ecological challenges; few are known to function in growth or other primary processes. 17-Hydroxygeranyllinalool diterpene glycosides (DTGs) are abundant herbivory-induced, structurally diverse and commonly malonylated defense metabolites in *Nicotiana attenuata* plants. By identifying and silencing a malonyltransferase, *NaMaT1,* involved in DTG malonylation, we found that DTG malonylation percentages are normally remarkably uniform, but when disrupted, result in DTG-dependent reduced floral style lengths, which in turn result from reduced stylar cell sizes, IAA contents, and YUC activity; phenotypes that could be restored by IAA supplementation or by silencing the DTG pathway. Moreover, the *Nicotiana* genus-specific JA-deficient short-style phenotype also results from alterations in DTG malonylation patterns. Decorations of plant specialized metabolites can be tuned to remarkably uniform levels, and this regulation plays a central but poorly understood role in controlling the development of specific plant parts, such as floral styles.
DOI: https://doi.org/10.7554/eLife.38611.001

**\*For correspondence:**
baldwin@ice.mpg.de

## Introduction

Malonylation is a ubiquitous modification of proteins and specialized metabolites. In plant specialized metabolism, the malonyl group is largely transferred from malonyl-coenzyme A (Malonyl-CoA) to the C'6 of the glycosyl moiety of glucoconjugates (*Taguchi et al., 2005*). The malonyl residue is thought to confer structural diversity, stability, and solubility to the decorated metabolites and provide a means of detoxifying xenobiotics (*Suzuki et al., 2002*; *Taguchi et al., 2005*; *Koirala et al., 2014*; *Suzuki et al., 2004*). Malonylation of anthocyanins enhances pigment stability and color intensity at the pH of the intracellular milieus (*Suzuki et al., 2002*). In addition, malonylated anthocyanins are preferentially transported to vacuoles in *Arabidopsis thaliana* (*Zhao et al., 2011*). The malonylation of glycosides is catalyzed by malonyltransferases, members of the biochemically versatile BAHD acyltransferase family (*D'Auria, 2006*). Although tens of malonyltransferases have been identified and functionally characterized in vitro (*Luo et al., 2007*; *Manjasetty et al., 2012*; *Bontpart et al., 2015*), their in vivo functions are largely unknown.

17-hydroxygeranyllinalool diterpene glycosides (DTGs) are abundant (mg/g FW) secondary metabolites in green tissues of many solanaceous plants, including tobacco (*Nicotiana* spp.), pepper (*Capsicum annuum*) and wolfberry (*Lycium chinense*) (*Jassbi et al., 2006*; *Lee et al., 2008*; *Heiling et al., 2010*). DTGs consist of a 17-hydroxygeranyllinalool aglycone that is decorated at the C3 and C17 hydroxyl positions by glucose, which in turn is modified by glucose, rhamnose and malonyl moieties, in various combinations (*Figure 1A* and *Figure 1—figure supplement 1*). So far,

**eLife digest** Plants produce tens of thousands of molecules called secondary metabolites that are thought to help them cope with threats from their environment, such as attack by insects or ultraviolet radiation from the sun. Wild coyote tobacco plants produce large amounts of a particular class of secondary metabolite known as DTGs. Insects feeding on tobacco plants containing DTGs cause less damage and produce fewer offspring

Plants modify many secondary metabolites by attaching tags known as malonyl groups to them. Enzymes called malonyl transferases take a malonyl group from another substrate and attach it to the secondary metabolite. This process can be repeated so that an individual secondary metabolite molecule may have many malonyl groups attached to it. Previous studies have shown that insects feeding on tobacco plants trigger more malonyl groups to be attached to DTGs, but it is not clear what effect this has on the plants.

To simulate attack by an insect, Li et al. punctured holes in the leaves of tobacco plants and applied saliva from tobacco hornworms. The experiments show that more DTGs modified with malonyl groups accumulated in these plants compared to untreated plants. However, there was no change in the average number of malonyl groups added to individual DTGs during the modification process. Further experiments show that a malonyl transferase enzyme called NaMaT1 adds malonyl groups to DTGs in coyote tobacco plants. The flowers of plants that produce less of this protein have shorter styles (a tube structure that guides pollen to the egg cells at the base of the flower) and are less fertile than flowers in normal plants.

These experiments demonstrate that, along with helping plants to defend themselves from herbivores, DTGs regulate how flowers grow and develop. It was generally thought that secondary metabolites do not play important roles in how plants grow when they are not under stress. Indeed, plant breeders frequently select crops that produce lower levels of secondary metabolites in order to increase their nutritional value. Therefore, the findings of Li et al. may help improve the outcomes of crop breeding programs.

DOI: https://doi.org/10.7554/eLife.38611.002

46 different DTGs (21 chemical formulas and several structural isomers) have been characterized in *Nicotiana attenuata*, an ecological model plant with a rich portfolio of specialized metabolites; however, the glycoside Lyciumoside IV and its malonylated products, Nicotianoside I and Nicotianoside II, constitute more than 80% of the DTG pool (*Poreddy et al., 2015*). In *N. attenuata*, DTGs function in resistance against the specialist herbivore, tobacco hornworm (*Manduca sexta*) (*Lou and Baldwin, 2003*; *Jassbi et al., 2008*; *Heiling et al., 2010*). The malonylated DTGs are particularly strongly induced by *M. sexta* feeding and jasmonate signaling (*Heiling et al., 2010*). The malonyl moieties of DTGs are lost from the DTGs soon after their ingestion by *M. sexta* larvae due to the alkaline environment of *M. sexta* oral secretions and midgut (*Poreddy et al., 2015*). This observation rules out a central role for malonylation of DTGs in antiherbivore defense, and suggests that other arenas need to be explored for potential functions mediated by this malonylation.

Specialized metabolites are primarily thought to function in mediating an organism's ecological interactions, to help optimize Darwinian fitness. Many are produced in response to particular ecological interactions, such as those that are induced in response to specific attackers. However, many secondary metabolites have effects on growth and development which are more specific than those that might result from resource trade-offs between growth and putative defense metabolites (*Züst and Agrawal, 2017*). For example, the insect feeding-induced glucosinolate breakdown product, indole-3-carbinol, arrests growth by interacting with the auxin receptor Transport Inhibitor Response (TIR1) as an auxin antagonist (*Katz et al., 2015*). Arabidopsis glucosinolates are also thought to modulate plant biomass, flowering time and the circadian clock, and inhibit root growth (*Kerwin et al., 2011*; *Jensen et al., 2015*; *Francisco et al., 2016*; *Malinovsky et al., 2017*). Hyperaccumulation of flavonoids is associated with stunted growth and developmental abnormalities, which are assumed to result from effects on auxin transport (*Franke et al., 2002*; *Bonawitz et al., 2014*; *Steenackers et al., 2017*). With the exception of the glucosinolates and flavonoids, little is

known about these potential 'primary' roles for other branches of specialized (also known as 'secondary') metabolism.

The jasmonate signaling pathway is undoubtedly among the most important in plant defense responses to herbivory, regulating a large portion of herbivory-responsive specialized metabolism, including DTGs in *N. attenuata* (*Kessler et al., 2004*; *Kallenbach et al., 2012*; *Li et al., 2017*; *Li et al., 2018*). However, jasmonates also regulate root growth, anther dehiscence, male fertility, fruit ripening and senescence (*Staswick et al., 1992*; *Xie et al., 1998*; *Wasternack et al., 2013*; *Stitz et al., 2014*). Jasmonate signaling-deficient genotypes of *N. attenuata*, including RNAi lines targeting ALLENE OXIDE CYCLASE (irAOC) and CORONATINE INSENSITIVE 1 (irCOI1), and ectopic expression of Arabidopsis JASMONIC ACID METHYL TRANSFERASE (JMT) and simultaneous silencing of METHYL JASMONATE ESTERASE (MJE) (JMT/mje) which dramatically decreased JA signaling, are reported to have short styles, resulting in reduced fertility (*Stitz et al., 2014*). Transgenic *N. tabacum* plants deficient in COI1 also display short styles (*Wang et al., 2014*). However, the mechanism of this seemingly *Nicotiana* genus-specific jasmonate-regulated phenotype is completely obscure.

Auxin plays a pivotal role throughout the entire lifespan of a plant, particularly in determination of floral development. It can influence cell division, cell expansion and cell differentiation, and thereby regulate a wide spectrum of developmental processes (*Benjamins and Scheres, 2008*). Mutants of *AtPIN1*, a polar auxin transporter, developed naked, pin-shaped inflorescences and abnormalities in all flower parts, confirming that auxin signaling contributes to flower development (*Okada, 1991*). Consistent with this, mutants of the auxin biosynthesis genes TRYPTOPHAN AMI NOTRANSFERASE OF ARABIDOPSIS (TAA) and YUCCA display severe defects in floral patterning, and complete sterility (*Cheng et al., 2006*; *Stepanova et al., 2008*). Auxin can also regulate gynoecium morphogenesis, including style elongation, through a concentration gradient from the apical to the basal part of the gynoecium (*Nemhauser et al., 2000*).

In this study, we discovered that the malonylation percentage of DTGs is remarkably uniform across development, treatments, and tissue types, and identified the gene involved in this malonylation. We ask whether this uniformity is essential for plant development. Silencing this gene caused strikingly and specifically short styles, and this phenomenon vanished when silencing this gene in DTG-deficient plants. We analyzed phytohormone levels, enzymatic activity, and effects of exogenous application, and all evidence indicated that the short style phenotype is caused by the influence of DTG malonylation status on IAA biosynthesis. Finally, we illustrate that the *Nicotiana* genus-specific JA-deficient style phenotype is caused by disturbing DTG malonylation patterns. Our work demonstrates that abnormal JA signaling could dysregulate DTG malonylation patterns, thereby affecting plant style development via auxin signaling.

## Results

### DTG malonylation percentages show remarkable uniformity across herbivory elicitation, developmental stages and tissues

To easily describe and understand the malonylation of DTGs, *N. attenuata* DTGs were classified, based on how many malonyl moieties they contained, into four categories: core, monomalonylated, dimalonylated, and trimalonylated (*Figure 1A* and *Figure 1—figure supplement 1*). In addition, we used a formula to calculate DTG malonylation percentage in each sample, based on numbers of malonyl moieties in each compound (*Figure 1D*). As previously reported, the biosynthesis of DTGs, especially malonylated DTGs, is strongly induced by mimicking *M. sexta* larval feeding (*Figure 1B*, [*Lou and Baldwin, 2003*; *Jassbi et al., 2008*; *Heiling et al., 2010*]). Remarkably, although all types of malonylated DTGs increased after *M. sexta* elicitation, there was no difference in malonylation percentage between treatment and control. To further analyze DTG malonylation patterns over plant development, DTGs were analyzed in flower buds of different stages. Total DTGs decreased dramatically over flower development, whereas the malonylation percentage was very stable (*Figure 1C and F*). By mining previously published metabolite data sets from different plant tissues (*Li et al., 2016*), we found that although different malonylated DTGs are highly variable across different tissues, with coefficients of variation (CV) ranging from 64 to 107 (*Figure 1—figure supplement 2B–*

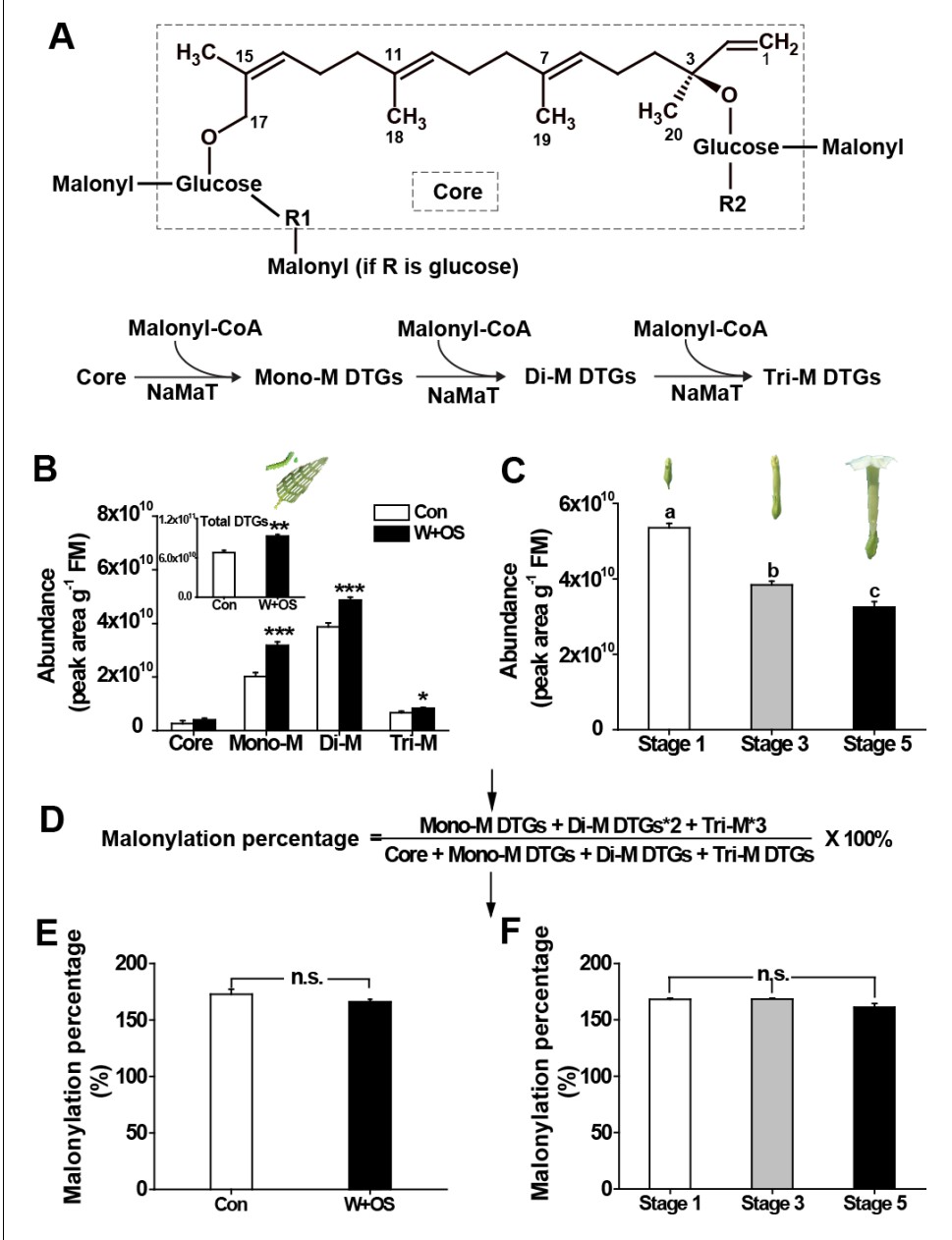

**Figure 1.** The malonylation percentage of DTGs is highly uniform across treatments and developmental stages. (A) Structures of 17-DTGs and their malonylation options. R1 could be glucose, rhamnose or hydrogen. R2 could be rhamnose or hydrogen. NaMaT indicates malonyltransferase in *N. attenuata*. (B) Relative abundance (mean + SE; $n = 5 – 6$) of different malonylated DTGs in *N. attenuata* leaves, after herbivory elicitation by immediately applying *M. sexta* oral secretions to freshly created leaf puncture wounds (W + OS), in comparison to the same nodal positions of plants without any treatment (Con). Inset: Total DTG abundance in control and W + OS elicited leaves. (C) Total DTGs in different stages of floral development. The formula (D) describes how malonylation percentage was calculated. (E, F) Malonylation percentage (mean + SE; $n = 5 – 6$) of DTGs in leaves treated by W + OS (D) or different developmental flower stages (E). Asterisks indicate significant differences between controls and treatments (*p<0.05; ***p<0.001; Student's *t*-tests). Different letters indicate significant differences among floral developmental stages (p<0.05, one-way ANOVA followed by Tukey's HSD *post-hoc* tests).

DOI: https://doi.org/10.7554/eLife.38611.003

The following source data and figure supplements are available for figure 1:

**Source data 1.** DTG profiles and malonylation percentages under different treatments and developmental stages.

*Figure 1 continued on next page*

*Figure 1 continued*

DOI: https://doi.org/10.7554/eLife.38611.006
**Source data 2.** DTG profiles and malonylation percentages across different tissues.
DOI: https://doi.org/10.7554/eLife.38611.007
**Figure supplement 1.** Structural diversity of *N. attenuata* DTGs, their diagnostic molecular ion [M + Na]$^+$ and retention times.
DOI: https://doi.org/10.7554/eLife.38611.004
**Figure supplement 2.** The DTG malonylation percentage is remarkably uniform across different tissues.
DOI: https://doi.org/10.7554/eLife.38611.005

*F*), the malonylation percentage was more uniform, with a CV of only 11: a significant outlier (*Figure 1—figure supplement 2G*).

## NaMaT is responsible for DTG malonylation

Manipulating gene(s) controlling the malonylation process is the most straightforward way to disentangle the function of DTG malonylation and its remarkably uniformity. To do this, *N. attenuata* malonyltransferase (MaT) genes that have high similarity with *NtMaT1* (*Taguchi et al., 2005*), were used to conduct a phylogenetic analysis with functionally characterized MaTs (*Figure 2A* and *Supplementary file 1*). There are five putative MaTs aligned in the same clade with other MaTs of the *Nicotiana* genus. Among these MaTs, NIATv7_g22417 and NIATv7_g34586 shared the highest protein sequence identity with NtMaT1: 91.2% and 91.4%, respectively; followed by NIATv7_g13429, which shared 80.6% protein sequence identity with NtMaT1 (*Figure 2—figure supplement 1*). Protein sequence alignment showed that four of the candidate MaTs contained the two conserved BAHD enzyme motifs HXXXDG and DFGWG, but not NIATv7_g21823, which had only a HXXXDG motif near the protein's center portion (*Figure 2—figure supplement 1A*). Notably, two putative MaTs, NIATv7_ g39356 and g21823, which share the highest identity with NtMaT1, also contained the flavonoid acyltransferase conserved motif, YFGNC (*Figure 2—figure supplement 1A*), indicating that these two MaTs are homologues of NtMaT1.

As gene co-expression network analysis is a powerful way to predict functions of unknown genes (*Serin et al., 2016*; *Higashi and Saito, 2013*), we performed a cluster analysis of *N. attenuata* MaTs with known DTG biosynthesis genes, using published multi-tissue and -treatment RNA-seq data (*Brockmöller et al., 2017*) (*Supplementary file 2*). Among the five putative NaMaTs, three were within the same clade as known DTG biosynthesis genes (*Figure 2B*). To determine whether any of these candidates were able to transfer the malonyl moiety from Malonyl-CoA to DTGs, we purified the most abundant core DTG, Lyciumoside IV (*Poreddy et al., 2015*), from *N. attenuata* leaves and performed in vitro enzyme activity assays using purified recombinant GST-tagged NaMaTs. Consistent with the expression patterns, the in vitro recombinant enzyme activity show that the same three genes: NaMaT1, NaMaT2 and NaMaT3 could catalyze malonylation from Lyciumoside IV to its monomalonylated form, Nicotianoside I, together with a comparatively minor production of the dimalonylated form, Nicotianoside II (*Figure 2C*).

To determine whether these three *N. attenuata* MaTs transcripts responded to *M. sexta* feeding as would be expected from the induced dynamics of DTGs, transcript abundance data were extracted from previously published RNA-seq data from leaves attacked by *M. sexta* larvae (*Ling et al., 2015*). *NaMaT1* was strongly induced within 5 hr after the onset of *M. sexta* larval feeding (59-fold), and then slightly decreased by 9 hr (*Figure 2D*). In contrast, both NaMaT2 and NaMaT3 transcripts were suppressed by *M. sexta* feeding, which is opposite to the dynamics of the *M. sexta*-induced DTG profile (*Figure 2D* and *Figure 1B*).

## Silencing *NaMaT1* affects floral development

To investigate the function of *N. attenuata* MaTs, virus-induced gene silence (VIGS) was used to silence all three candidate genes. The tobacco rattle virus VIGS vector migrates to growing meristems and thus efficiently silences target genes in new tissues in all parts of plants (*Galis et al., 2013*). VIGS of the carotenoid biosynthetic gene *N. attenuata phytoene desaturase* (*NaPDS*), which causes photobleaching where the gene is silenced, demonstrated efficient silencing in floral tissues (*Figure 3—figure supplement 1A*). Possibly because of low transcript levels (*Figure 2D*), neither

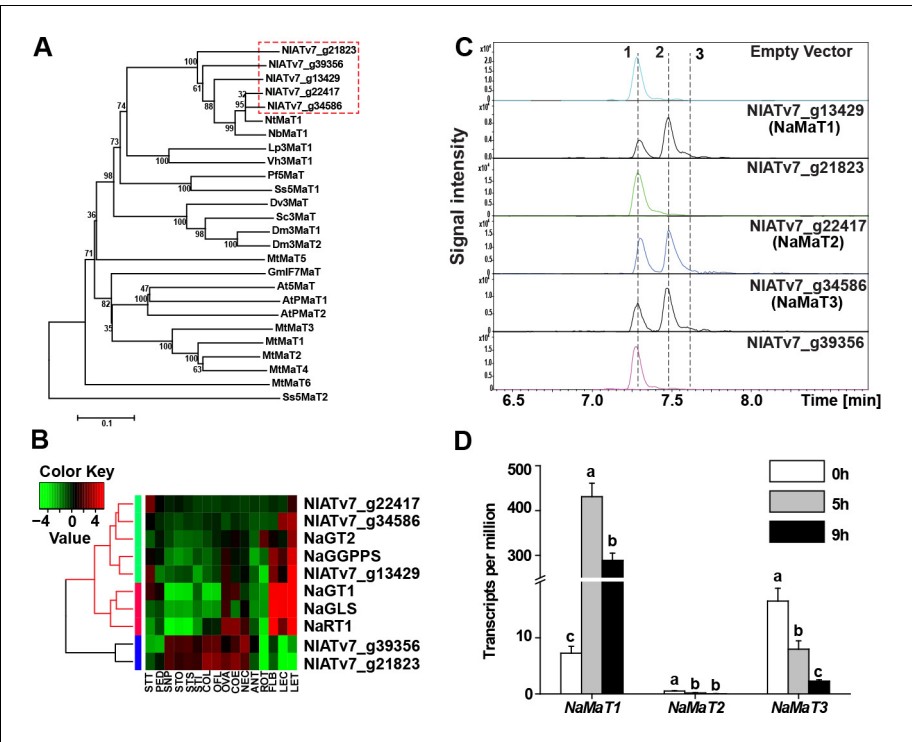

**Figure 2.** NaMaT1 mediates DTG malonylation in *N. attenuata*. (A) Phylogenetic analysis of potential *N. attenuata* malonyltransferases (genes in red dotted box) and functionally characterized malonyltransferases in other species by amino acid sequence with accession number shown in *Supplementary file 1*. (B) Heatmap representing the expression of malonyltransferases and reported DTG biosynthetic genes in *N. attenuata*. LET, leaf treated (25 hr after wounding and elicitation with *M. sexta* oral secretions); LEC, leaf control; STT, stem treated; PED, pedicels; SNP, style without pollination; STO, style outcrossed; STS, style selfed; STI, stigma; COL, corolla late; OFL, opening flower; OVA, ovary; COE, corolla early; NEC, nectaries; ANT, anthers; ROT, root OS-elicited; FLB, flower bud. (C) Extracted ion chromatograms of m/z 271.2420, corresponding to the DTG aglycone, of in vitro assay products of recombinant malonyltransferases. Peaks 1, 2 and 3 were identified as Lyciumoside IV, Nicotianoside I and Nicotianoside II, respectively. (D) *NaMaT1*, *NaMaT2* and *NaMaT3* transcript counts (mean + SE; *n* = 3) were analyzed from RNAseq data of *M. sexta*-attacked leaves at indicated time points. Different letters indicate significant differences among treated time points (p<0.05, one ANOVA followed by Tukey's HSD *post-hoc* tests).
DOI: https://doi.org/10.7554/eLife.38611.008

The following source data and figure supplement are available for figure 2:

**Source data 1.** Relative transcript abundance of DTG biosynthesis genes in *N. attenuata*.
DOI: https://doi.org/10.7554/eLife.38611.010

**Figure supplement 1.** Analyzing protein sequences of NbMaT1, NtMaT1 and malonyltransferases in *N. attenuata*.
DOI: https://doi.org/10.7554/eLife.38611.009

NaMaT2 nor NaMaT3 were successfully silenced (data not shown), and so further work focused on NaMaT1. The transcript abundance of *NaMaT1* in VIGS-NaMaT1 plants (VIGS-MaT1) decreased more than 80% compared with VIGS controls (empty vector, VIGS-EV), without affecting the abundance of *NaMaT2* or *NaMaT3* transcripts (*Figure 3—figure supplement 1B*). Notably, we also analyzed transcript abundance of two other reported DTG biosynthesis genes, geranylgeranyl diphosphate synthase (*NaGGPPS*) (*Jassbi et al., 2008*) and geranyllinalool synthase (*NaGLS*) (*Falara et al., 2014*) in VIGS plants, and found that *NaGLS* transcript abundance significantly increased in VIGS-MaT1 leaves (*Figure 3—figure supplement 1C*).

NaMaT1 VIGS plants show similar overall growth phenotypes as control plants, including the morphology of shoots, leaves, the floral exterior, corolla limb and stamen (*Figure 3—figure supplement 1D*). However, we observed that VIGS-MaT1 plants rarely produced capsules. This observation was quantified by counting capsule numbers at the end of seed set, which showed that VIGS-MaT1 plants produced on average only one capsule every two plants (*Figure 3—figure supplement 2A*).

In a second experiment, we determined that styles of VIGS-MaT1 plants were extremely short, less than half the length of VIGS-EV styles (*Figure 3A and E*). To elucidate whether a decrease of cell number or cell length caused the short style phenotype, we visualized the style cells from freshly opening flowers using a histochemical stain specific for callose. The stylar cell length was strongly reduced, whereas cell number was not (*Figure 3B–D*). In order to clarify whether the short-style phenotype is due to alteration of DTG malonylation, or to other unknown functions of NaMaT1, VIGS was conducted on DTG-deficient plants, irGGPPS. The irGGPPS stably transformed line is specifically silenced in the expression of one of three *GGPPSs* in the *N. attenuata* genome: the enzyme that controls the flux of substrates into the DTG pathway, and irGGPPS produces only ca. 10 – 15% of the DTGs levels of WT plants (*Heiling et al., 2010*). The results show that the short-style phenotype vanished in irGGPPS-background VIGS-MaT1 plants (*Figure 3*), although *NaMaT1* was silenced to a similar degree in the styles of irGGPPS and EV plants (*Figure 3—figure supplement 3A*).

To determine whether the sterility of VIGS-MaT1 plants was due to the physical separation of stigma and anthers, or additional effects on the function of male or female parts, we conducted hand-pollinations of VIGS plants. Pollen from either VIGS-MaT1 or VIGS-EV plants applied to the VIGS-EV pistil produced normal capsules and similar numbers of seeds in each capsule (*Figure 3—figure supplement 2B and C*). However, only withered capsules resulted from hand-pollination of VIGS-MaT1 pistils with VIGS-EV pollen, and no capsules resulted from pollination of VIGS-MaT1 pistils with VIGS-MaT1 pollen. The withered capsules produced dramatically fewer seeds than those of VIGS-EV capsules, although the seed germination rate was not significantly affected (*Figure 3—figure supplement 2C and D*).

## NaMaT1 affect DTGs, JAs and IAA

To determine whether NaMaT1 controls the induced malonylation of DTGs *in planta*, we measured DTGs in the leaves of VIGS plants following MeJA treatment, which is known to strongly induce DTG biosynthesis and malonylation. Silencing *NaMaT1* significantly reduced the malonylation percentage, primarily through an increase in DTGs with low malonylation degree (core and monomalonylated DTGs), in comparison to VIGS-EV controls (*Figure 4A*). This effect was more pronounced after leaves were treated with MeJA (*Figure 4B*). In order to elucidate the short-style phenotype, we analyzed the DTG profile in styles and inflorescences, which are hypothesized to affect the early stages of style development (*Smyth, 1990*; *Yanofsky, 1995*). In addition to increasing core and monomalonylated DTGs, dimalonylated and trimalonylated DTGs significantly decreased in inflorescences and styles with stigmas, leading to a dramatic decrease in malonylation percentage in VIGS-MaT1 in comparison to VIGS-EV (*Figure 4C,D*): all individual core and monomalonyated DTGs were dramatically increased in VIGS-MaT1 styles, whereas trimalonylated DTGs were no longer detectable (*Figure 4—figure supplement 1A*). Because the irGGPPS background could rescue the VIGS-MaT1 short style phenotype, we analyzed DTG profiles in the inflorescence of VIGS plants in the irGGPPS and EV backgrounds. Similarly, as for VIGS of EV plants, the malonylation percentage in *NaMaT1*-silenced irGGPPS plants significantly decreased, but not as much as in *NaMaT1*-silenced EV plants (*Figure 4—figure supplement 1B*; 26% reduction for EV and 16% reduction for irGGPPS). The intermediate decrease in malonylation percentage for irGGPPS results from both an increase of core and monomalonylated DTGs, and a decrease of dimalonylated and trimalonylated DTGs; overall, DTG levels are much lower in the irGGPPS background (*Figure 4—figure supplement 1B and C*).

Disturbed phytohormone levels frequently result in serious floral phenotypes (*Okada, 1991*; *Stitz et al., 2014*). We analyzed JA, JA-Ile and auxin levels in leaves, inflorescences and styles (*Figure 4*). In both control and MeJA-treated leaves, JAs and IAA in VIGS-MaT1 were similar to those of controls. VIGS of *NaMaT1* reduced JA and JA-Ile levels significantly in inflorescences compared with the VIGS-EV, without affecting IAA contents (*Figure 4C*). In contrast, VIGS of *NaMaT1* reduced IAA levels in styles and stigmas by 40%, but had no significant effect on JAs levels (*Figure 4D*). Because there is a sophisticated tissue-specific regulatory mechanism for auxin biosynthesis and homeostasis (*Ljung et al., 2001*), and to gain insight into why silencing *NaMaT1* specifically affected IAA in styles, we compared the IAA levels among different flower tissues and first stem (S1) leaves. Styles and stigmas contained the highest levels of IAA among all the measured tissues, about 6.5-fold more than that of the S1 leaf (*Figure 4—figure supplement 2B*). Additionally, styles and stigmas also contain relatively large amounts of DTGs (*Figure 4—figure supplement 2C*).

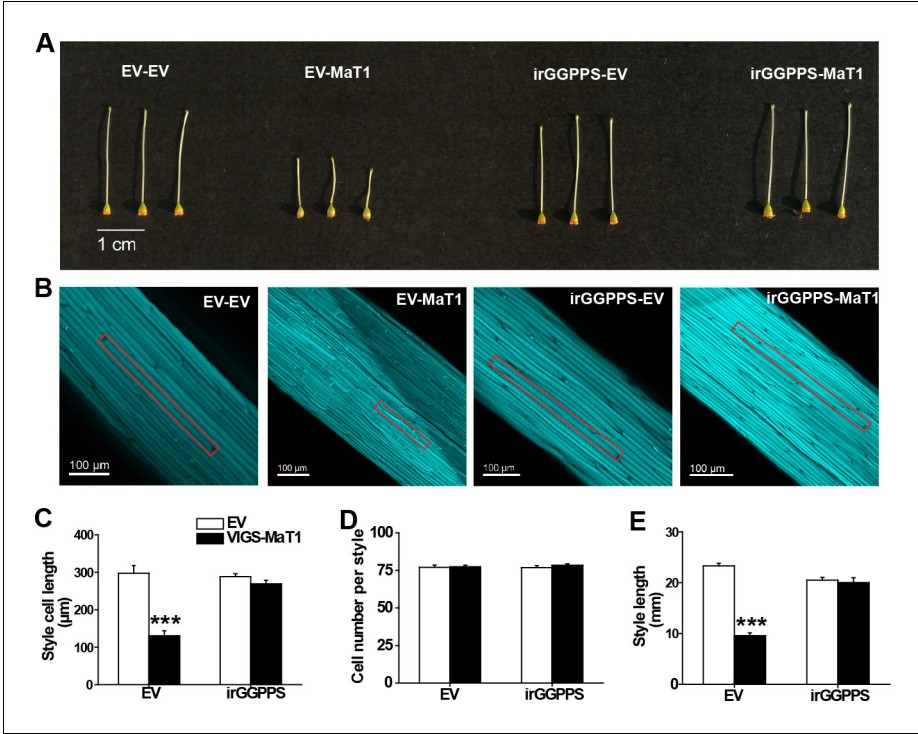

**Figure 3.** Silencing *NaMaT1* expression dramatically shortens style lengths by decreasing cell size rather than cell number in a GGPPS-dependent manner. VIGS of NaMaT1 (VIGS-MaT1) versus a control VIGS empty vector (VIGS-EV) were infiltrated in EV and irGGPPS stably-transformed plants, referred to respectively as EV-EV, EV-MaT1, irGGPPS-EV and irGGPPS-MaT1. (**A**) Representative photographs of gynoecia from opened flowers are shown from the indicated genotypes. (**B**) Typical aniline blue stained style samples from indicated genotypes. Representative cells are highlighted with a vermilion rectangle. Mean (+ SE; $n = 5 – 7$) cell length (**C**) and cell number per style (**D**) of styles of opened flowers of indicated plants were measured with fluorescence microscopy. (**E**) Style lengths (+ SE; $n = 5$) were measured from flowers on the first day of opening. Asterisks indicate significant differences between VIGS-EV and VIGS-MaT1 (***$p < 0.001$; Student's *t*-test).
DOI: https://doi.org/10.7554/eLife.38611.011

The following source data and figure supplements are available for figure 3:

**Source data 1.** Silencing *NaMaT1* expression dramatically shortens style lengths by decreasing cell size rather than cell number in a GGPPS-dependent manner.
DOI: https://doi.org/10.7554/eLife.38611.015
**Source data 2.** Effect of silencing *NaMaT1* on gene expression in leaves.
DOI: https://doi.org/10.7554/eLife.38611.016
**Source data 3.** VIGS of *NaMaT1* affects the fertility of *N. attenuata* plants.
DOI: https://doi.org/10.7554/eLife.38611.017
**Source data 4.** VIGS of *NaMaT*1 in both irGGPPS and irAOC backgrounds did not affect *NaYUC-like 2* transcript accumulation in styles.
DOI: https://doi.org/10.7554/eLife.38611.018
**Figure supplement 1.** Effect of silencing *NaMaT1* on gene expression in leaves, and plant morphology.
DOI: https://doi.org/10.7554/eLife.38611.012
**Figure supplement 2.** VIGS of *NaMaT1* affects the fertility of *N. attenuata* plants.
DOI: https://doi.org/10.7554/eLife.38611.013
**Figure supplement 3.** VIGS of *NaMaT*1 in both irGGPPS and irAOC backgrounds did not affect *NaYUC-like 2* transcript accumulation in styles.
DOI: https://doi.org/10.7554/eLife.38611.014

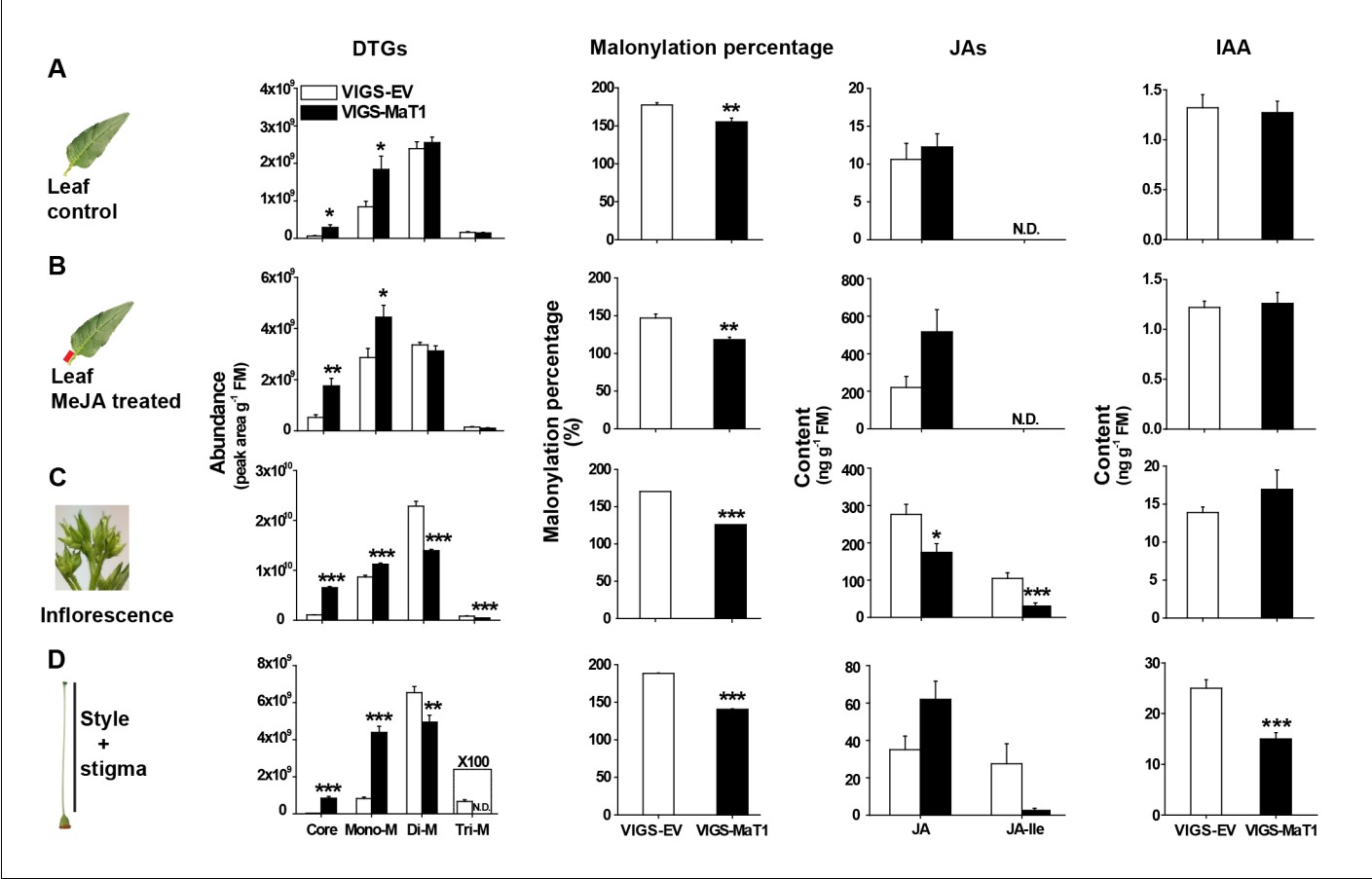

**Figure 4.** Silencing *NaMaT1* influences DTG profiles and malonylation percentages across all tissues, but alters JAs and IAA specifically in flowers. S1 leaves from early elongated VIGS plants were treated with 20 µL lanolin paste (A, leaf control) or 150 µg MeJA in lanolin paste (B, leaf MeJA treated) and samples were harvested after 3 days. The inflorescence (C), style and stigma (D) were harvested from flowering plants. Relative abundance of different malonylated DTGs (first column) were analyzed from upper samples. Malonylation percentage (second column) was calculated based on the DTG data using the formula (*Figure 1*). Jasmonates (JAs, third column) and IAA (fourth column) were analyzed from the same samples as those used for DTG quantification. N.D. indicates compounds which were not detected because of low concentrations. Asterisks above each column indicate significant differences between EV and VIGS-MaT1 plants (*p<0.05; **p<0.01; ***p<0.001; Student's *t*-test).
DOI: https://doi.org/10.7554/eLife.38611.019

The following source data and figure supplements are available for figure 4:

**Source data 1.** Silencing *NaMaT1* influences DTG profiles and malonylation percentages across all tissues, but alters JAs and IAA specifically in flowers.
DOI: https://doi.org/10.7554/eLife.38611.022
**Source data 2.** DTG profiles in VIGS plants having irGGPPS, irAOC, or EV backgrounds.
DOI: https://doi.org/10.7554/eLife.38611.023
**Source data 3.** IAA and DTGs accumulate in high concentrations in the gynoecium.
DOI: https://doi.org/10.7554/eLife.38611.024
**Figure supplement 1.** DTG profiles in VIGS plants having irGGPPS, irAOC, or EV backgrounds.
DOI: https://doi.org/10.7554/eLife.38611.020
**Figure supplement 2.** IAA and DTGs accumulate in high concentrations in the gynoecium.
DOI: https://doi.org/10.7554/eLife.38611.021

## Decreased IAA levels are responsible for the VIGS-MaT1 short-style phenotype

To determine whether decreased auxin caused the short style phenotype, we analyzed IAA and its precursor tryptophan (Trp) in styles of both EV and irGGPPS plants inoculated with VIGS-MaT1. While IAA decreased in VIGS-MaT1 of EV plants, Trp levels increased dramatically (*Figure 5A*). These differences were eliminated when we silenced *NaMaT1* in the irGGPPS background

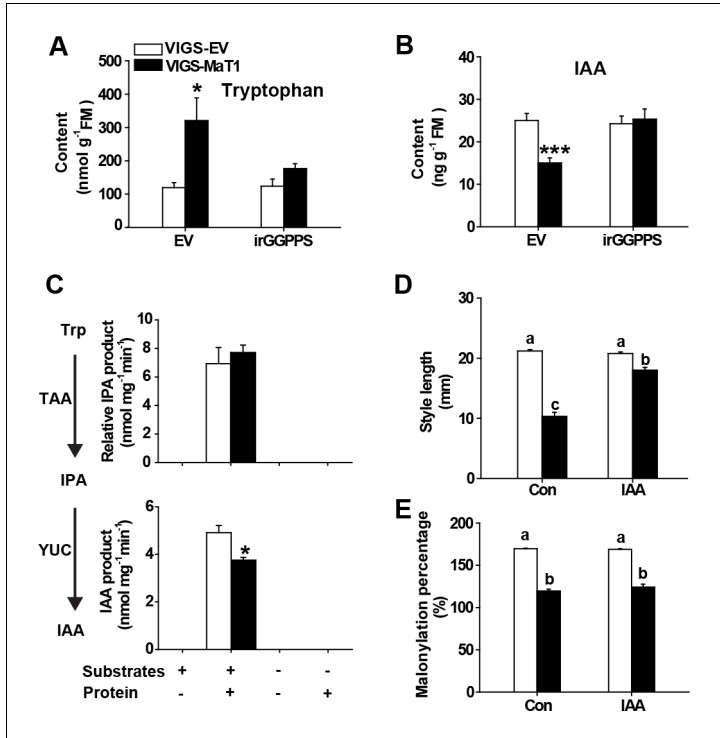

**Figure 5.** The short styles of VIGS-MaT1 flowers are associated with reduced IAA biosynthesis. Tryptophan (**A**) and IAA (**B**) levels (mean + SE; $n = 6$) were analyzed from style samples dissected from flowers one day before anthesis. Crude protein was extracted from the same set of samples and then used for in vitro IAA biosynthetic enzyme activity assay. Relative in vitro enzyme assay product abundance (mean + SE; $n = 5$) of IPA (**C**, top panel) and IAA (**C**, bottom panel), respectively. (**D**) Style length (mean + SE; $n = 10$) of flowers injected with 0.1% DMSO (Con) or 0.1% DMSO together with 10 µM IAA 2 days before anthesis. (**E**) Malonylation percentage (mean + SE; $n = 6$) was calculated from DTGs from the same set of samples. Asterisks indicate significant differences between VIGS-EV and VIGS-MaT1 plants (*p<0.05; ***p<0.001; Student's *t*-test). Different letters indicate significant differences among different lines or treatments (p<0.05, one-way ANOVA followed by Tukey's HSD *post-hoc* tests).

DOI: https://doi.org/10.7554/eLife.38611.025

The following source data and figure supplements are available for figure 5:

**Source data 1.** The short styles of VIGS-MaT1 flowers are associated with reduced IAA biosynthesis.
DOI: https://doi.org/10.7554/eLife.38611.028

**Source data 2.** Relative abundance of flavonoids in *NaMaT1* and/or *NaGGPPS* silenced plants.
DOI: https://doi.org/10.7554/eLife.38611.029

**Source data 3.** Brassinosteroid does not contribute to the VIGS-MaT1 short style.
DOI: https://doi.org/10.7554/eLife.38611.030

**Figure supplement 1.** Silencing *NaMaT1* and *NaGGPPS* does not affect flavonoids in different tissues, except that VIGS of NaMaT1 reduces rutin in the style.
DOI: https://doi.org/10.7554/eLife.38611.026

**Figure supplement 2.** Brassinosteroid does not contribute to the VIGS-MaT1 short style.
DOI: https://doi.org/10.7554/eLife.38611.027

---

(*Figure 5A and B*), which again indicated that the effect of *NaMaT1* on IAA biosynthesis in styles is DTG-dependent. To elucidate the effect of *NaMaT1* on IAA biosynthesis, we extracted crude protein from EV and VIGS-MaT1 styles and performed in vitro enzyme activity assays of IAA biosynthesis. Surprisingly, NaTAA1 activity was similar between EV and VIGS-MaT1. However, the transformation from IPA to IAA, which is thought to be catalyzed by YUCCA, was significantly impaired in VIGS-MaT1 style (*Figure 5C*). We then analyzed the transcript abundance of *YUCCA-like* genes in *N. attenuata* by RNAseq, and found one *YUCCA-like* gene, *YUC-like* 2 (*Machado et al., 2016*), to be highly expressed in styles (*Supplementary file 2*). The transcript abundance of *YUC-like 2* was similar

between VIGS-MaT1 and VIGS-EV in both backgrounds of irGGPPS and EV plants (*Figure 3—figure supplement 3B*). Exogenous application IAA approximately restored the short styles to their normal lengths (*Figure 5D*), without affecting the malonylation degree of stylar DTGs (*Figure 5E*), results which are consistent with the hypothesis that the truncated style resulted from decreased IAA biosynthesis.

The effect of flavonoids on auxin transport has been well characterized (*Peer and Murphy, 2007*), and flavonoids also could be used as substrate by the homolog of NaMaT1 in *N. tabacum*, NtMaT1 (*Taguchi et al., 2005*). To test whether the effect of VIGS-MaT1 on auxin could be caused by changes in flavonoid metabolism, we measured the major flavonoids in *N. attenuata* plants. We found that both leaves and styles contained similar levels of kaempferol-3-*O*-glucoside and kaempferol-3-*O*- rhamnosyl glucoside between VIGS-MaT1 and VIGS-EV, and only rutin was significantly decreased in VIGS-MaT1 style compared with EV styles (*Figure 5—figure supplement 1A and B*). Furthermore, the stably transformed line irGGPPS also displays flavonoid levels indistinguishable from EV in both leaves and inflorescences, consistent with previous data showing that irGGPPS plants have similar rutin contents as WT plants in both greenhouse and field studies (*Heiling et al., 2010*).

Brassinosteroids and the transcription factor style2.1 were reported to promote cell elongation and thereby affect style length in *Primula* spp. (primroses) and in *Solanum lycopersicum* (tomato), respectively (*Chen et al., 2007*; *Huu et al., 2016*). We identified the homolog of the *S. lycopersicum style2.1* gene in *N. attenuata*, but both RNAseq and qRT-PCR failed to detect transcripts in *N. attenuata* styles (*Supplementary file 2*). PveCYP734A50 was reported to degrade brassinosteroids and thereby control style length in *Primula veris* (*Huu et al., 2016*). Through searching the *N. attenuata* genome database, we found the three closest homologs of *PveCYP734A50*. RT-PCR analysis of those three genes in styles showed that the transcript abundance of one gene, *NIATv7_g25593*, matched very well with the short style phenotype, being most abundant in styles after VIGS-MaT1 of EV, and having low abundance in styles after VIGS-EV or VIGS-MaT1 of irGGPPS plants (*Figure 5—figure supplement 2B*). We then designed a specific construct and silenced *NIATv7_g25593* using VIGS, but did not observe any effect on style length (data not shown). To test whether brassinosteroids may contribute to the VIGS-MaT1 style phenotype in other ways, we exogenously applied brassinolide to VIGS-MaT1 flower buds. This treatment did not recover the short style phenotype of VIGS-MaT1 (*Figure 5—figure supplement 2D*). Thus, we can rule out the possibility of those two mechanisms contributing to the VIGS-MaT1 short style phenotype.

## The *N. attenuata* JA-deficient short-style phenotype is caused by disturbed DTG malonylation patterns

The stylar phenotype of VIGS-MaT1 plants was strongly reminiscent of the short styles of plants with JA signaling deficiencies, which is only reported in *Nicotiana* species, as far as we know. Among species for which JA-deficient phenotypes have been reported, only the genus *Nicotiana* produces DTGs (*Heiling et al., 2016*). Therefore, we hypothesized that the short styles of JA-deficient plants results from disturbed DTG malonylation patterns, similar to those of VIGS-MaT1 plants. To address this hypothesis, we analyzed DTG profiles using a stably transformed *N. attenuata* irAOC line as a severely jasmonate-deficient model. In both herbivore-damaged and control leaves, malonylation percentages in irAOC were significantly higher than in EV plants (*Figure 6—figure supplement 1A*). In flower buds, the irAOC malonylation percentage was also significantly higher than in EV (*Figure 6—figure supplement 1B*). MeJA treatment could partially restore the irAOC short style phenotype (*Figure 6—figure supplement 1C* and [*Stitz et al., 2014*]), and also partially restored the malonylation percentages towards EV levels (*Figure 6—figure supplement 1D*). In line with leaves and flower buds, the malonylation percentage of irAOC styles was also significantly higher than in EV (*Figure 6A*). To further test this hypothesis, we compared style lengths in VIGS-MaT1 and VIGS-EV of both EV and irAOC plants. Silencing *NaMaT1* in EV or irAOC plants resulted in similarly truncated styles (*Figure 6B* and *Figure 3—figure supplement 3C*). The lack of an additive effect suggests that irAOC short styles may result from the same mechanism as is responsible for the short styles of VIGS-MaT1 EV plants. Consistently, the malonylation percentages after VIGS-MaT1 in both EV and irAOC backgrounds were similar, much lower than the VIGS-EV of both EV and irAOC genetic backgrounds (*Figure 6C*).This reduction is mainly caused by an increase of core and monomalonylated DTGs and a decrease of trimalonylated DTGs (*Figure 4—figure supplement 1D and*

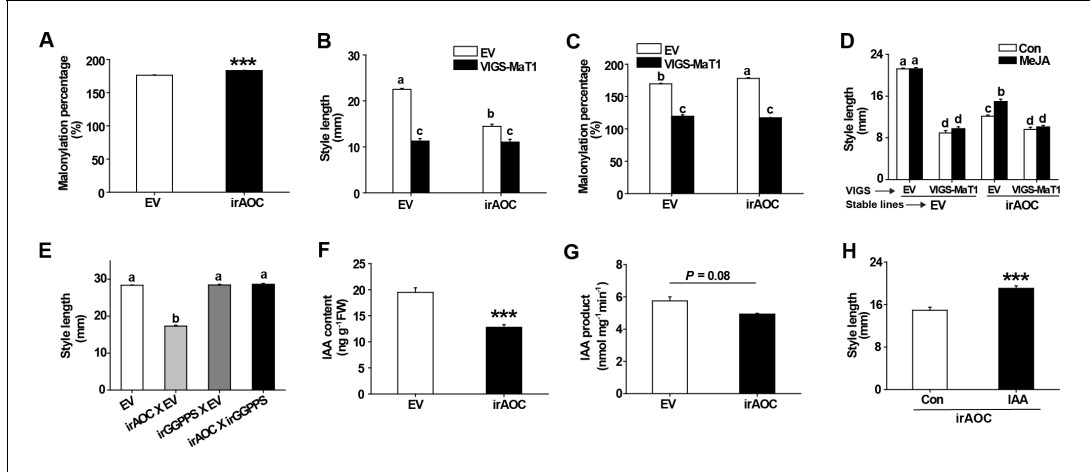

**Figure 6.** The short styles of jasmonate-deficient irAOC plants result from disturbed DTG malonylation status. (A) DTG malonylation percentage (mean + SE; *n* = 5) was calculated based on DTGs of styles dissected from flowers one day before anthesis. (B) Style lengths (mean + SE; *n* = 10) were measured from freshly opened flowers of VIGS-EV and VIGS-MaT1 silencing in the background of two stably transformed lines, EV or irAOC, as indicated. (C) Malonylation percentages (mean + SE, *n* = 3) were calculated based on DTGs in styles dissected from flowers one day before anthesis of the four genotypes as described in (B). (D) Mean style lengths (mean + SE; *n* = 10) were measured from freshly opened flowers of the four genotypes, which had been treated with lanolin paste with MeJA (MeJA) or only lanolin paste as control (Con), 2 days previously. (E) Style lengths (mean + SE; *n* = 5) were measured from freshly opened flowers of the four genotypes, which were from crosses of EV or irAOC with EV or irGGPPS. (F) IAA levels (mean + SE; *n* = 5) were analyzed from the same samples as in (A). (G) IAA (mean + SE; n = 3) was quantified from in vitro enzyme activity assay products, in which the enzyme activity is from crude protein extracted from styles as in (A). (H) Style lengths (mean + SE; *n* = 10 – 16) of freshly opened irAOC flowers which had been injected with 0.1% DMSO aqueous solutions with 10 µM IAA (IAA) or only 0.1% DMSO aqueous solution as controls (Con) 2 days before. Asterisks indicate significant differences between different genotypes or treatments (***p<0.001; Student's *t*-test). Different letters indicate significant differences among different plant lines or treatments (p<0.05, one-way ANOVA followed by Tukey's HSD *post-hoc* tests).

DOI: https://doi.org/10.7554/eLife.38611.031

The following source data and figure supplement are available for figure 6:

**Source data 1.** The short styles of jasmonate-deficient irAOC plants result from disturbed DTG malonylation status.

DOI: https://doi.org/10.7554/eLife.38611.033

**Source data 2.** DTG malonylation status and style length are affected by silencing a JA biosynthesis gene (AOC), and are partly recovered by exogenous applications of MeJA.

DOI: https://doi.org/10.7554/eLife.38611.034

**Figure supplement 1.** DTG malonylation status and style length are affected by silencing a JA biosynthesis gene (AOC), and are partly recovered by exogenous applications of MeJA.

DOI: https://doi.org/10.7554/eLife.38611.032

*E*). Furthermore, MeJA treatment could partially restore irAOC-EV short styles, but not the short styles of VIGS-MaT1 in irAOC or EV plants (*Figure 6D*). These data revealed that NaMaT1 functions downstream of NaAOC to control style length. Importantly, when we crossed irAOC with irGGPPS to silence both DTG and JA production, the irAOC short-style phenotype was completely restored to normal style lengths of EV or WT plants (*Figure 6E*).

Because the VIGS-MaT1 short style phenotype results from attenuated IAA levels, we measured IAA contents in irAOC styles. IAA levels in irAOC styles were much lower than in EV (*Figure 6F*). The protein activity responsible for transforming IPA to IAA was marginally decreased in irAOC styles compared to that in EV (*Figure 6G*), but the transcript abundance of *NaYUC-like 2* was similar between VIGS-MaT1 and VIGS-EV in the backgrounds of irAOC and EV (*Figure 3—figure supplement 3D*). Finally, exogenous applications of IAA partially restored the truncated styles of irAOC plants to WT lengths (*Figure 6H*). These results were consistent with the hypothesis that the short styles of JA-deficient *N. attenuata* are also caused by disturbed DTG malonylation patterns.

## Discussion

Specialized, or secondary metabolites are usually thought only to mediate plant responses to specific environmental conditions, and not to be directly involved in plant growth, development and reproduction – the providence of primary metabolism. Here, we report that the decoration of specialized metabolites demonstrated remarkable uniformity across development, tissues, and treatments, and found that one malonyltransferase (MaT) contributes significantly to this process. Silencing this gene disturbed 17-hydroxygeranyllinalool diterpene glycoside (DTG) malonylation patterns, and stunted elongation of flower styles during development and fertility by affecting auxin biosynthesis. Abnormal jasmonate signaling could also disturb DTG malonylation patterns, causing similar stylar developmental defects.

Enzyme assays showed that three MaT proteins, NaMaT1-3, catalyze the first step of DTG malonylation, from Lyciumoside IV to Nicotianoside I, and the second step, from Nicotianoside I to Nicotianoside II, in vitro (*Figure 2C*). As Lyciumoside IV only has two glucose moieties (*Figure 1—figure supplement 1*) and the abundances of tri-glucosylated core DTGs are too low to purify from plants, we were unable to test the ability of those NaMaTs to catalyze the third malonylation reaction. However, in vivo silencing of *NaMaT1* dramatically decreased dimalonylated and trimalonylated DTGs in inflorescences and styles (*Figure 4*), which indicates that NaMaT1 might also control the third step of the malonylation reaction *in planta*. Notably, in vivo silencing of *NaMaT1* changed individual DTG abundances in complex ways. For example, although the total abundance of dimalonylated DTGs decreased significantly in *NaMaT1*-silenced styles, the abundance of one dimalonylated DTG, Nicotianoside X, did not change significantly, whereas the abundance of two dimalonylated compounds, Nicotianoside XII and Nicotianoside VII, increased (*Figure 4–Figure supplement 1AFigure 4—figure supplement 1A*). Currently, we cannot rule out the possibility that some of these changes are not directly mediated by NaMaT1 catalytic activity, but result from systemic influence, such as substrate limitation. Future work should test NaMaT1 substrate specificity for all DTGs, when technology can support the purification of low-abundance components in sufficient amounts. Based on our current results, we infer that NaMaT1 is a malonyl-CoA:DTG malonyltransferase, which may catalyze three consecutive malonylation steps, as reported for Dm3MaT2, which catalyzes consecutive dimalonyl transfers to anthocyanin (*Suzuki et al., 2004*). Notably, two characterized MaTs in the *Nicotiana* genus, NbMaT1 and NtMaT1, accept aromatic glycosides as substrate (*Taguchi et al., 2005*; *Liu et al., 2017*). Phylogenic and protein identity analysis showed that NbMaT1 and NtMaT1 share the highest sequence similarity with NaMaT2 and NaMaT3, about 90%, whereas the identity with NaMaT1 is 80% (*Figure 2A* and *Figure 2—figure supplement 1B*). Moreover, cross-tissue comprehensive nontargeted metabolomics analyses did not reveal any malonylated flavonoids in *N. attenuata* (*Li et al., 2016*), although *NaMaT1* is highly expressed in *M. sexta*-infested leaves. Thus, we assume that NaMaT1 mainly functions in DTG malonylation *in planta*, and NaMaT2 and NaMaT3 could also accept aromatic glycosides as substrates, like NbMaT1 and NtMaT1.

Specialized metabolites are thought to be strongly regulated by environmental cues, and are often thought to function in a plant's adaptive responses to environmental changes. Here, our data revealed remarkable uniformity of DTG malonylation status, although individual DTGs are plastic. For the regulation of DTG biosynthesis, feed-back and feed-forward regulation are both described by Heiling et al. (in preparation): silencing a rhamnosyltransferase gene inhibits the transcript accumulation of all upstream genes, and silencing *NaGGPPS* also suppresses transcript accumulation of downstream biosynthetic genes. Here, we show that VIGS of *NaMaT1* tended to increase both *NaMaT2* and *NaMaT3* expression, and significantly increased *NaGLS* expression (*Figure 3—figure supplement 1B and C*). Two or more elements oppositely controlling one process is a powerful homeostatic strategy, as is found in JA-Ile conjugation and JA-Ile hydroxylation which maintains JA-Ile homeostasis (*Kang et al., 2006*; *Woldemariam et al., 2012*). Here, the induction of *NaMaT1* and suppression of *NaMaT2* and *NaMaT3* in response to *M. sexta* feeding may represent a similar mechanism to control DTG malonylation percentage during the response to herbivore attack (*Figure 2D*). Using feed-back and -forward regulation together with differential expression levels of *NaMaTs*, plants maintain DTG malonylation percentage within a very narrow range. This is likely important because a malonylation percentage alteration as low as 4.1% in irAOC is associated with a drastically shortened style and infertility (*Figure 6A*). Additionally, the very low variation in the malonylation percentages reported here throughout the results indicates that DTG malonylation is precisely

regulated. Although long-term, high-dose MeJA treatments of the leaves of VIGS plants (3 days using the same amount of MeJA normally applied to glasshouse-grown plants, with leaves typically 3-5x the size of the leaves of VIGS plants) reduced DTG malonylation percentages (VIGS-EV, *Figure 4A and B*), the locally transient endogenous JA bursts elicited by *M. sexta* larval feeding did not alter malonylation percentages (*Figure 1E*). Importantly, both high (irAOC) and low (VIGS-MaT1) malonylation percentages were accompanied by truncated styles. Notably, malonylation percentage decreased in irGGPPS-MaT1 inflorescences but without apparently affecting style length of irGGPPS-MaT1 plants (*Figure 4—figure supplement 1B*). We think that this is because irGGPPS plants produce much lower levels of DTGs, and thus the total abundance of DTGs in inflorescences may not be enough to affect style development, regardless of their malonylation.

Induction of the jasmonate signaling pathway, including via MeJA treatment and *M. sexta* attack, strongly induced *NaMaT1* expression (*Figure 2D*) and DTG malonylation (*Figure 1B*), indicating that NaMaT1 is responsible for the increase in malonylated DTGs. However, as noted above, high-dose MeJA treatment suppressed malonylation percentages, and silencing JA biosynthesis (irAOC) increased malonylation percentage in many tissues (*Figure 6A* and *Figure 6—figure supplement 1A and B*). This apparent contradiction may be explained by the possibility that other MaTs also contribute to the DTG malonylation process, as indicated by the fact that *M. sexta* feeding suppresses NaMaT2 and NaMaT3 (*Figure 2D*). Although *NaMaT1* transcript accumulation decreased more than 80% in VIGS-MaT1 leaves (*Figure 3—figure supplement 1B*), the malonylation percentage was only reduced by 19.6% (*Figure 4B*), again consistent with the hypotheses that other MaTs catalyze DTG malonylation. Future research could test NaMaT2 and NaMaT3 functions by manipulating their expression with more robust means, such as CRISPR-mediated genome editing.

Our experiments shed light on this phenomenon by demonstrating that the DTG biosynthesis specific NaGGPPS is required for the development of the truncated style phenotype (*Figure 3*), and this truncated style phenotype results from the low auxin levels found in the styles, which in turn are due to reduced IAA biosynthesis (*Figure 5*). We hypothesize that this style-specific effect occurs because the style synthesizes large amounts of DTGs (*Figure 4—figure supplement 2C*), and VIGS-MaT1 results in a stronger decrease in malonylation percentage in styles than it does in leaves (*Figure 4B*). In addition, styles contained the highest levels of IAA of all tissues analyzed (*Figure 4—figure supplement 2*) and are likely the auxin source for at least the gynoecium, as predicted by the apical-basal gradient of auxin theory (*Nemhauser et al., 2000*). Our data from irGGPPS plants with depleted DTGs supports our hypothesis that the truncated style of VIGS-MaT1 is related to DTGs, but the exact molecular events that lead to this phenomenon remain unknown. Comparing individual DTG patterns in VIGS plants in the irGGPPS, irAOC and EV backgrounds showed that changes of individual compounds are complex. As we do not know how plants perceive DTGs, it is very challenging to figure out which individual compound or combination of individual compounds could be perceived by plants and affect YUC activity. The low abundance and structural similarity of many DTGs, along with the instability of malonylation in solution ex vivo, make purification difficult (*Heiling et al., 2016*), and for the malonylated DTGs it is not feasible to obtain sufficient amounts or sufficient stability for the establishment and duration of bioassays. Furthermore, given the huge number of combinations of candidate DTGs, a gain-of-function test using specific DTGs is currently prohibitive. For example, if we were to try one or two DTGs at once, not accounting for isomers and limiting ourselves to the 16 DTG chemical formulae for which we could present relative quantification here, the total number of possible combinations is $2.8 \times 10^{25}$. This number of course could be restricted based on comparisons of the dynamics of individual compounds to the dynamics in malonylation percentage of styles in different treatment groups (EV, irAOC, irGGPPS; VIGS-EV vs. VIGS-MaT1), although the possibility that a ratio rather than a single compound is required could still result in many combinations that require testing.

The biosynthesis and malonylation of DTGs, which are diterpenoid derivatives, may interact with other compounds from this pathway, especially some phytohormones, like strigolactones, abscisic acid, cytokinins, gibberellins and brassinosteroids (*Cazzonelli and Pogson, 2010*). Notably, flavonoids were thus far the only known secondary metabolites that could inhibit auxin transport, resulting in stunted growth in *Arabidopsis*, *Medicago*, tomato and apple (*Brown et al., 2001*; *Eckardt, 2006*; *Besseau et al., 2007*; *Schijlen et al., 2007*; *Dare et al., 2013*), and malonyl-CoA is a required substrate for flavonoid biosynthesis (*Kreuzaler and Hahlbrock, 1975*). It is possible that silencing *NaMaT1* may increase flavonoid biosynthesis, thereby affecting auxin and causing the short

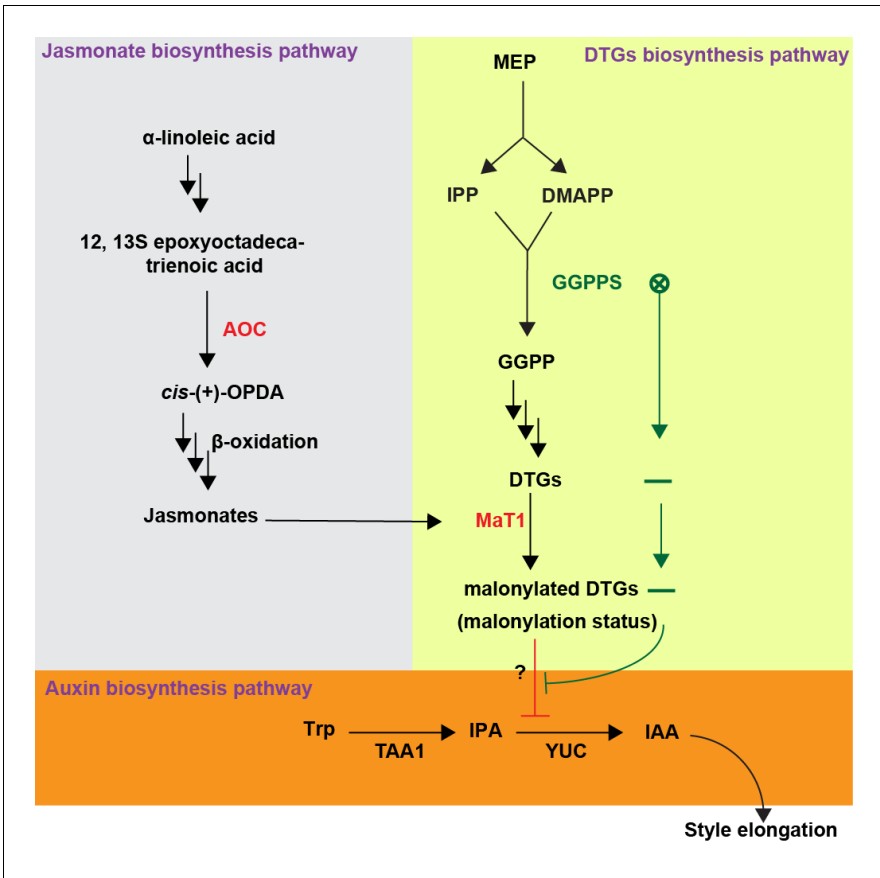

**Figure 7.** Model summarizing how DTGs malonylation affects style elongation of *N. attenuata*. Geranylgeranyl diphosphate synthase (GGPPS) forms GGPP from the MEP pathway products IPP and DMAPP. This GGPP is used for DTG biosynthesis and a certain portion of these DTGs are decorated with malonic acid moieties by NaMAT1. When AOC or MaT1 are silenced (red), this disturbs the uniformity of DTG malonylation, resulting in compounds which inhibit the rate-limiting step of IAA biosynthesis, YUC catalytic activity. When GGPPS is silenced (green), leading to DTG deficiency, the suppression of YUC catalytic activity is alleviated. MEP, Methylerythritol 4-phosphate pathway; IPP, isopentenyl diphosphate; DMAPP, dimethylallyl diphosphate.
DOI: https://doi.org/10.7554/eLife.38611.035

style phenotype. However, flavonoids did not significantly increase in VIGS-MaT1 plants (*Figure 5—figure supplement 1A and B*), and levels of rutin even decreased. Rutin has been reported not to affect auxin transport (*Jacobs and Rubery, 1988*). In summary, the short style phenotype is a malonylation-dependent phenotype that is based specifically on DTGs. Bioassay-driven metabolite extractions from VIGS-MaT1 tissues may help to discover the specific compound(s) inhibiting YUC activity.

In conclusion, this study reveals the uniformity of specialized metabolite malonylation and the importance of this uniformity: when this uniformity is disturbed by silencing a malonyltransferase gene and its regulator, JA signaling, the cells of the tissue which accumulates the highest levels of DTGs fail to elongate normally, resulting in a stunted style phenotype (*Figure 7*). This stunted style phenotype resulting from either JA signaling deficiencies or silencing *NaMaT1*, results from the inhibition of auxin biosynthesis. Elucidating how plants achieve this finely tuned DTG malonylation status, and which specific DTG structure is responsible for the inhibition of auxin biosynthesis in styles are exciting goals for future research. At a functional level, by shortening styles by titrating the degree of malonyl DTG decorations, plants may be able to regulate their outcrossing rates in response to environmental factors that influence JA signaling or DTGs accumulation, such as herbivory rates.

# Materials and methods

## Key resources table

| Reagent type (species) or resource | Designation | Source or reference | Identifiers | Additional information |
|---|---|---|---|---|
| Gene (*Nicotiana attenuata*) | *NaMaT1* | PRJNA355166 | XM_019403695.1 | |
| Gene (*N. attenuata*) | *NaMaT2* | PRJNA355166 | XR_002066055.1 | |
| Gene (*N. attenuata*) | *NaMaT3* | PRJNA355166 | XM_019382488.1 | |
| Gene (*N. attenuata*) | *NIATv7_g21823* | Nicotiana attenuata Data Hub | NIATv7_g21823 | |
| Gene (*N. attenuata*) | *NIATv7_g39356* | Nicotiana attenuata Data Hub | NIATv7_g39356 | |
| Strain, strain background (*Agrobacterium tumefaciens*) | GV1301 | DOI: 10.1007/978-1-62703-278-0_9 | GV1301 | |
| Strain, strain background (*Escherichia coli*) | BL21 (DE3) | New England Biolabs inc. | catalog#: C2527I | |
| Genetic reagent (*N. attenuata*) | Empty vector (EV) | DOI: 10.1371/journal.pone.0001543 | EV | |
| Genetic reagent (*N. attenuata*) | irGGPPS | DOI: 10.1105/tpc.109.071449 | irGGPPS | |
| Genetic reagent (*N. attenuata*) | irAOC | DOI: 10.1073/pnas.1200363109 | irAOC | |
| Transfected construct (tobacco rattle virus) | pBINTRA6/pTV00 | DOI: 10.1007/978-1-62703-278-0_9 | pBINTRA6/pTV00 | |
| Recombinant DNA reagent | Gateway vector pDEST15 | Invitrogen | catalog#: 11802014 | |
| Sequence-based reagent | Oligonucleotides | Sigma-aldrich | | Supplied in *Supplementary file 3* |
| Peptide, recombinant protein | GST-NaMaT1 | This paper | | |
| Peptide, recombinant protein | GST-NaMaT2 | This paper | | |
| Peptide, recombinant protein | GST-NaMaT3 | This paper | | |
| Peptide, recombinant protein | GST-NIATv7_g21823 | This paper | | |
| Peptide, recombinant protein | GST-NIATv7_g39356 | This paper | | |

*Continued on next page*

*Continued*

| Reagent type (species) or resource | Designation | Source or reference | Identifiers | Additional information |
|---|---|---|---|---|
| Commercial assay or kit | SuperScript First-Strand Synthesis System for RT-PCR | Invitrogen | catalog#: 11904018 | |
| Chemical compound, drug | TRIzol ™ Reagent | Invitrogen | catalog#: 15596026 | |
| Chemical compound, drug | Indole-3-pyruvic acid | Sigma-aldrich | CAS:392-12-1 | |
| Chemical compound, drug | Glutathione-Sepharose 4B | GE Healthcare | GE17-0756-01 | |
| Software, algorithm | MEGA6 | MEGA | http://www.megasoftware.net/ | |
| Software, algorithm | SPSS statistic 17.0 | SPPS inc. | http://www-01.ibm.com/software/analytics/spss/ | |

## Plant material and growth conditions

The 31 st inbred generation of *N. attenuata* originating from a collection at the DI ranch in southwestern Utah USA was used as the wild-type background for all transformants. Previously described homozygotes of the third transformed generation of irGGPPS (A-07-230-5) (*Heiling et al., 2010*), irAOC (A-07-457-1) (*Kallenbach et al., 2012*), and an empty vector control line (EV, A-03-009-1) (*Schwachtje et al., 2008*) were used. Seeds were germinated on a mixture of plant agar with Gamborg's B5 medium in sterile petri dishes and seedlings were transferred to pots and grown under 19 − 35°C, 16 hr light (supplemental lighting by Philips Sun-T Agro 400W and 600W sodium lights) and 60 − 65% relative humidity as previously described (*Krügel et al., 2002*; *Saedler and Baldwin, 2004*). The VIGS plants were obtained following the procedures described in (*Galis et al., 2013*). Briefly, leaves of young rosette-stage plants were pressure infiltrated with a mixture of *Agrobacterium tumefaciens* containing pBINTRA and either pTV-MaT1 or pTV00 (EV control). VIGS experiments were repeated at least three times.

## Plant treatment and sample collections

For methyl jasmonate (MeJA) treatments, leaf petioles of rosette-stage plants were treated with 20 μL lanolin paste containing 150 μg MeJA (Sigma-Aldrich), or with 20 μL of pure lanolin as a control. MeJA treatment of flower buds followed a similar protocol except the lanolin paste volume was 10 μL and applied to the pedicel of each bud. For *Manduca sexta* W + OS elicitations, rosette-stage leaves were wounded with a pattern wheel, and 20 μL of diluted *M. sexta* regurgitant (1:5 in distilled water) was gently rubbed into the freshly created puncture wounds using a clean gloved finger as previously described (*Schittko et al., 2001*). Three days after the treatment, leaves, excluding the midvein, were harvested for metabolite and RNA extraction. For flower samples, flowers were harvested following standardized developmental stages (*Li et al., 2017*). For floral tissue samples, five to ten opening flowers from one plant were dissected and each tissue was separately pooled to create one biological replicate. Five replicates were used for each tissue type. For style samples used for metabolites, phytohormones and in vitro enzyme activity assays, ten flower buds were harvested one day before anthesis, dissected and styles were pooled as one biological replicate. All style length measurements were conducted on first-day open flowers.

## DTGs and flavonoid analysis by UPLC-Q-TOF

Samples were ground in liquid nitrogen and aliquoted to 10 – 100 mg depending on the tissues and their known DTG and flavonoid concentrations (precise mass was recorded). Approximately 100 mg of aliquoted samples were extracted using 1 mL 80% methanol aqueous buffer and analyzed on a micrOTOF-Q II system (Bruker Daltonics) as previously described (*Heiling et al., 2016*; *Li et al., 2016*). QuantAnalysis (Bruker Daltonics) software was used to integrate the DTG peak areas based on each compound's diagnostic m/z value and retention time as described in *Figure 1—figure supplement 1*.

## In vitro recombinant protein assays

Malonyltransferase candidate genes that had high similarity to *NtMaT1* were identified from the *N. attenuata* data hub (http://nadh.ice.mpg.de/). Fragments with coding regions of candidate malonyltransferases were amplified from cDNA with gene specific primers as described in *Supplementary file 3*. Full-length cDNAs of the candidate genes without stop codons were introduced to Gateway destination vector pDEST™ 15, through entry vector pENTR™ according to the manufacturer's instructions. The recombinant proteins were expressed in *E. coli* BL21 (DE3), extracted and purified using Glutathione-Sepharose 6B (GE Healthcare) in accordance with the manufacturer's instructions. The in vitro enzyme activity assays were conducted as previously described (*Taguchi et al., 2010*). Briefly, 1 µg purified protein was added to the reaction mixture (50 µL), which was 50 mM potassium phosphate buffer (pH 8.0) with 200 µM malonyl-CoA, 5 mM β-mercaptoethanol, and 8 µg DTGs. The mixture was incubated at 30°C for 1 hr, and the reactions were stopped by adding 10 µL 1M HCl. Methanol (40 µL) was then added to the reaction mixture, and this was subsequently used for DTGs quantification by UPLC-Q-TOF as described above.

## Imaging pistils

The complete pistils were harvested, fixed (ethanol: acetic acid, 3:1) and stained with aniline blue as previously described (*Mori et al., 2006*). Stained samples were viewed under a confocal laser scanning microscope (LSM 880, Zeiss, Jena, Germany) in channel mode with a 20x objective (Plan-Apochromat 20x/0.8) and a 405 nm laser diode for illumination. Excitation, emission and detection windows were set via a 405 nm main beam splitter and the QUASAR detector range between 480 and 550 nm, respectively. The pinhole size was 41 µm, while the Z-stack step size was 1 µm. Tiled Z stacks were acquired to obtain all necessary details. After image acquisition, the scanned tiles were stitched in ZEN (black 2012, Zeiss, Inc.). Representative images were obtained with maximum intensity projections in ImageJ 1.50e. Cell lengths were measured in ImageJ 1.50e.

## Gene expression analysis

For quantitative RT-PCR, RNA was extracted from 30 mg well-ground tissue using TRIzol reagent (Invitrogen), and the RNA quantity was confirmed by the 260/280 nm absorbance ratio using Nano-Drop™ (ThermoFisher Scientific). One µg RNA was used for reverse transcription by First strand cDNA synthesis kit (ThermoFisher Scientific). RT-qPCR was done via Mastermix (Eurogentec) SYBR Green reaction in a Stratagene 500 MX3005P Real-time qPCR machine. The primers used for mRNA detection of target genes by RT-PCR are listed in *Supplementary file 3*. The amplification specificity of primers was confirmed by single peaks in a dissociation curve following qPCR. The *N. attenuata IF5a-2* mRNA was used as internal control. RNA-seq data for gene expression in all *N. attenuata* tissues was previously published in NCBI with accession number PRJNA317743 (*Brockmöller et al., 2017*). To readily visualize these data in heatmaps, the raw data was transformed by log2. RNA-seq data for gene expression after *M. sexta* larval feeding was previously published in NCBI with accession number PRJNA223344 (*Ling et al., 2015*).

## Phytohormone quantifications

Jasmonates, IAA and tryptophan were measured as previously described (*Schäfer et al., 2016*). Briefly, aliquots of ca. 100 mg (precise mass recorded) frozen powdered samples were extracted with 800 µL extraction buffer containing the internal standards (20 ng $D_6$-JA, 20 ng $D_6$-JA-Ile, 3 ng $D_5$-IAA), purified by successive HR-X and HR-XC SPE column chromatography (MACHEREY-NAGEL), and finally analyzed on a EVO-Q Elite™ Triple quadrupole-MS (Bruker Daltonics). Prior to SPE

purification, 2 µL of the initial extract was diluted into 98 µL aqueous solution containing 255 fmol µL$^{-1}$ $^{13}C_9$, $^{15}N_1$-phenylalanine as an internal standard for the quantification of tryptophan.

## In vitro TAA1 and YUC enzyme activity assay

Approximately 20 mg frozen powdered samples were extracted with 200 µL cold extraction buffer (0.1M Tris-C1, pH 7.6; 5% polyvinylpolypyrrolidone; 2 mg/mL phenylthiourea; 5 mg/mL diethyldithio-carbamate; 0.05 M Na$_2$EDTA). The reaction for TAA1 enzyme activity assays was incubated at 55°C for 20 min, and other procedures were as previously described (*Tao et al., 2008*). The YUC enzyme activity assay was conducted as previously described (*Mashiguchi et al., 2011*). Both reactions were stopped by acidification with 5 µL 3M phosphoric acid, then adding 500 ng D$_5$-IAA as an internal standard before the reaction product was extracted three times with an equal volume of ethyl acetate. The supernatant was dried and the pellet was resuspended in 50 µL of methanol. The methanol-solubilized extracts were analyzed by UHPLC-MS (impact II, Bruker Daltonics) in negative ESI mode.

## Statistical analyses

All ANOVAs were performed in SPSS statistic 17.0 (SPSS Inc, http://www-01.ibm.com/software/analytics/spss/). The Student's *t*-tests were performed in Microsoft Office Excel 2010. Homogeneity of variance was evaluated in SPSS using Levene's test, and outliers were assessed by the function of Explor in SPSS with default parameters. The protein sequences were aligned by CLUSTAL W and phylogenetic trees were constructed using the maximum-likelihood method in MEGA6 (http://www.megasoftware.net/).

## Acknowledgements

We thank the gardening staff at the Max Planck institute for Chemical Ecology and the service group of the Department of Molecular Ecology. We thank Ran Li, Sven Heiling, Ming Wang, Lucas Cortes Llorca, Dapeng Li and Youngsung Joo for technical support and fruitful discussions. This work was supported by the Max Planck Society, and the European Research Council Advanced Grant (293926) Clockwork Green (to ITB).

## Additional information

### Competing interests

Ian T Baldwin: Senior editor, *eLife*. The other authors declare that no competing interests exist.

### Funding

| Funder | Grant reference number | Author |
|---|---|---|
| Deutsche Forschungsgemeinschaft | Collaborative Research Centre "ChemicalMediators in Complex Biosystems - ChemBioSys" (SFB 1127) | Ian T Baldwin Meredith C Schuman Rayko Halitschke |
| Max-Planck-Gesellschaft | Open-access funding | Ian T Baldwin |
| European Research Council | Senior Award: ClockWorkGreen 293926 | Ian T Baldwin |

The funders had no role in study design, data collection and interpretation, or the decision to submit the work for publication.

### Author contributions

Jiancai Li, Conceptualization, Data curation, Formal analysis, Validation, Investigation, Visualization, Methodology, Writing—original draft, Writing—review and editing; Meredith C Schuman, Data curation, Formal analysis, Visualization, Project administration, Writing—review and editing; Rayko Halitschke, Data curation, Visualization, Methodology, Project administration, Writing—review and

editing; Xiang Li, Validation, Investigation, Methodology; Han Guo, Austin Hammer, Validation, Investigation; Veit Grabe, Methodology; Ian T Baldwin, Conceptualization, Resources, Data curation, Supervision, Funding acquisition, Visualization, Writing—original draft, Project administration, Writing—review and editing

### Author ORCIDs
Jiancai Li (iD) http://orcid.org/0000-0002-4417-7612
Meredith C Schuman (iD) http://orcid.org/0000-0003-3159-3534
Rayko Halitschke (iD) http://orcid.org/0000-0002-1109-8782
Veit Grabe (iD) http://orcid.org/0000-0002-0736-2771
Ian T Baldwin (iD) http://orcid.org/0000-0001-5371-2974

### Decision letter and Author response
Decision letter https://doi.org/10.7554/eLife.38611.045
Author response https://doi.org/10.7554/eLife.38611.046

## Additional files

### Supplementary files
• Supplementary file 1. Gene accession numbers used in this study.
DOI: https://doi.org/10.7554/eLife.38611.036

• Supplementary file 2. The transcripts per million (TPM) values of DTGs biosynthesis pathway genes, *NaStyle2.1* and *NaYUC-like*, from RNAseq analysis.
DOI: https://doi.org/10.7554/eLife.38611.037

• Supplementary file 3. DNA primers used in this study.
DOI: https://doi.org/10.7554/eLife.38611.038

• Transparent reporting form
DOI: https://doi.org/10.7554/eLife.38611.039

### Data availability
All data generated or analysed during this study are included in the manuscript and supporting files

The following previously published datasets were used:

| Author(s) | Year | Dataset title | Dataset URL | Database, license, and accessibility information |
|---|---|---|---|---|
| Brockmoller T, Ling ZH, Li DP, Gaquerel E, Baldwin IT, Xu SQ | 2017 | Nicotiana attenuata Data Hub (NaDH): an integrative platform for exploring genomic, transcriptomic and metabolomic data in wild tobacco | https://www.ncbi.nlm.nih.gov/bioproject?term=PRJNA317743 | Publicly available at NCBI BioProject (accession no. PRJNA317743) |
| Ling ZH, Zhou WW, Baldwin IT, Xu SQ | 2015 | Insect herbivory elicits genome-wide alternative splicing responses in Nicotiana attenuata | https://www.ncbi.nlm.nih.gov/bioproject/?term=PRJNA223344 | Publicly available at NCBI BioProject (accession no. PRJNA223344) |

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
