## [Decision Letter]

Thank you for submitting your article "The decoration of specialized metabolites influences stylar development" for consideration by *eLife*. Your article has been reviewed by two peer reviewers, and the evaluation has been overseen by Joerg Bohlmann as the Reviewing Editor and Christian Hardtke as the Senior Editor. The reviewers have opted to remain anonymous.

The reviewers and the reviewing editor found that your manuscript makes an important and new contribution to plant biology. Both reviewers have a number of comments that may help improve the paper. The reviewers' comments are below. The two reviewers have discussed their individual reviews as part of our consultation session. The major request of both reviewers is for a more detailed characterisation of the effects of NaMaT1 silencing in the backgrounds silenced for other relevant biosynthesis genes.

*Reviewer #1:*

This manuscript describes the role of a group of specialized (secondary) plant metabolites for normal style development and plant fertility. The authors identify the major enzyme from *Nicotiana attenuata* that malonylates 17-hydroxygeranyllinalool diterpene glycosides (DTGs), a group of abundant, herbivore induced compounds in tobacco. Silencing this malonyl transferase NaMaT1 leads to reduced cell elongation in the style, and this effect appears to be mediated by a reduction in IAA biosynthesis. Importantly, it can be suppressed by silencing a key enzyme for DTG biosynthesis, arguing that the NaMaT1-VIGS effect is due to altered DTG malonylation levels. The authors also find that the short-style phenotype of JA-deficient plants that is observed in *Nicotiana*, but has not been described in other plants, is likely due to the influence of JA signaling on DTG malonylation levels.

Understanding how the vast diversity of plant specialized metabolites influences the plant's interactions with its environment and other organisms, but also the endogenous growth and developmental programs remains a key challenge in plant biology. In this respect, this paper provides exciting novel insight into the role of one class of specifically decorated metabolites. In particular, both the apparently strict control over the DTG malonylation levels throughout plant development and the highly specific effects of interfering with these malonylation levels highlight the functional importance of these compounds as putative novel regulators of phytohormone levels and plant growth. The manuscript thus makes an important contribution to our understanding of plant secondary metabolites.

The following three issues should be addressed to strengthen the manuscript further.

1) A key finding in this manuscript is that silencing NaMaT1 only results in short styles in a wild-type, but not in a NaGGPPS-silenced background. This experiment provides the main argument that the NaMaT1 effect is indeed mediated by the altered malonylation levels of DTGs, rather than some other undetected compound(s). However, it is not clear whether the style phenotype results from an excess accumulation of 'undermalonylated' DTGs, or from the altered malonylation percentage per se. It should be possible to address this by determining the malonylation percentage on the remaining DTGs in the VIGS-*Mat1* irGGPPS plants. If this is still reduced to the same extent as in VIGS-*Mat1* plants, it would argue that high levels of 'undermalonylated' DTGs interfere with auxin biosynthesis, rather than the malonylation level itself being critical.

2) The reason for the sterility of the VIGS-*Mat1* plants should be determined in more detail. Is it just the physical separation of stigma and anthers – in which case hand-pollination should rescue fertility – or is there also a functional difference in the styles that interferes with efficient pollen-tube growth or guidance, which might have interesting implications for plant incompatibility systems?

3) YUCCA activity appears to be reduced in VIGS-NaMaT1 styles, and the authors seem to favor an effect of altered DTG malonylation levels on YUCCA enzymatic activity, rather than on YUCCA gene expression. Are YUCCA mRNA levels altered in VIGS-NaMaT1 styles?

*Reviewer #2:*

Li and coworkers report on the importance of malonylated hydroxygeranylinalool diterpene glycosides (DTGs) in *Nicotiana attenuata*. They found that the malonylation percentage of DTGs was uniform in different organs of *N. attenuata* and after herbivore treatment. A correlation analysis lead them to the identification of three BAHD-acyltransferases (ATs), NaMaT1-3 able to catalyze a mono- and di- malonylation of the DTG, Lyciumoside IV. Virus induced gene silencing (VIGs) of NaMaT1 lead to a change in DTG malonylation patterns. A decrease in style length was the only phenotypic change for those NaMaT1-VIGs plants observed. Using plant hormone application as well as the background of knock-down plants (irGLS and irAOC) this phenotype was identified to be due to a decrease in auxin levels due to the inhibition of YUCCA. Furthermore the dependence of the effect on DTG and JA signaling could be shown.

Overall the paper tells a nice, relevant and interesting story of the complex regulation of plant metabolism and the linkage of primary and secondary metabolism. I do believe that the observed style phenotype in *N. attenuata* is dependent on JA signaling, DTGs and auxin and that the identified ATs are somewhat involved in DTGs malonylation. What the authors did not convince me of, yet, is that the phenotype is due to the malonylation percentage rather than key metabolites of the pathway described.

My main request is to see an analysis including individual DTGs instead of just pooling them into groups. Furthermore some sections require additional data or rephrasing.

- The authors look at DTGs based on their malonylation amounts. It would be of interest to look at single compounds. Could the observed effect of style length be due to single compounds rather than a shift in malonylation percentage? If this is not the case data should be provided to support that only the percentage changes. In fact one could test whether the percentage matters by shifting it through the application of DTG.

- The authors state that the phenotype is not observed with irGGPPS VIGS-MaT1 plants. No data however characterizing the chemo type of these plants is supplied. How do the DTGs levels compare to VIGS-MaT1 plants? The same goes for silencing efficiencies of MaT1. This holds for irAOC plants too. Please ensure that the reader has access to the full characterization: qRT for silencing,% of malonylation, mono-di-tri malonylation, style length, IAA and JA content, YUCCA activity for all the plants which are of importance to draw a conclusion. Please show whether YUCCA activity is inhibited in irGGPPS (+/- VIGS-MaT1) plants to support involvement of malonylated DTGs in this phenomenon. The authors claim that the mechanism is the same for irAOC plants. How do auxin levels look in those plants and is auxin biosynthesis (YUCCA) inhibited too?

- Expression level of MaT1 in VIGS plants was solely investigated in leaves the place of VIGS/*Agrobacterium* treatment. Those are however not the organs where a phenotype was observed. Explanations about the observation of a phenotype in styles, not the place of VIGS treatment should be included. The fifth paragraph of the Discussion states the style synthesises large amounts of DTGs. Why was no qRT done on styles? AT gene expression in irGGPPS and irAOC with EV and VIGS-MaT1 should be investigated in order to draw a full picture. Please answer the question of how knock down of MaT1 affects the plants in the background of irGGPPS and irAOC more carefully. For example, don't just show malonylation percentage but show at least mono-, di- and tri-malonylation. Especially as irAOC background changes malonylation percentage to be higher.

- The NaMaT1-3 enzymes mediate the mono- and di-malonylation of Lyciumoside IV. The mono-mannosylation however is unlikely to be the enzymes in vivo function as MaT1 VIGS plants produce greater amounts of the Nicotianoside I compared to EV plants. The diglycosylation, which the authors showed happens in vitro too, seems to be affected. Statements about enzyme function should be revisited in the paper as they sometimes seem rather blurry. Statements like "controls the last two malonylation steps" cannot be made as observed patterns can also be due to substrate limitations. In addition you cannot throw all the malonylated compounds together as one when talking about enzyme functions (unless you show that it really is one enzyme accepting those different substrates). Those compounds do differ greatly biochemically speaking. I would find it necessary to look at individual compounds rather than the pool of compounds categorized by mono-, di-, tri-malonylation, in order to figure out enzyme function. The specific enzyme function might help to prove the hypothesis about malonylation percentage or even show that the effect observed might be due to rather specific compounds. "found that one MaT mediates this process" is not true. Please phrase carefully. Regarding the three acyltransferases, the authors should provide details about similarity of genes/proteins mentioned in the first paragraph of the subsection “NaMaT is responsible for DTG malonylation”. Substrate specificity of NaMaT1 should be accessed especially due to observed decrease in rutin. You might not find malonylated flavonoids, but how about NaMaT1s ability to accept other acyl donors? Statements like those in the second paragraph of the Discussion do not mean anything unless tested with the protein in question. Please state how similar NaMaT1 and NbMaT1 are. Was NbMaT1 tested with DTGs?

- I do not find it surprising that the malonylation percentage of DTGs are uniform. From the biosynthesis perspective if you activate the pathway all compounds accumulate (in case of co-regulated genes) and this is what we see. In contrast, if the malonylation percentage of DTGs has the ability to regulate auxin would you not expect the percentage to vary in different organs or developmental stages rather than to be uniform? From my point of view it is necessary to look at individual compounds. Statements in the third paragraph of the Discussion about the uniformity of DTG malonylation status in contrast to strong regulation of specialist metabolites is rather strange. DTGs appear to be regulated after all as they are induced after herbivory.

---

## [Author Response]

We were able to conduct many new measurements using plants growing in our glasshouse, and existing samples. This allowed us to address most reviewer concerns. However, we do not yet have measurements for the DTG profile of irGGPPS VIGS-MaT1 styles. We agree with the reviewers that the short style phenotype we observe could be due to one or a few individual DTGs, and we agree that analyzing the chemical profile of irGGPPS VIGS-MaT1 styles is a possible way to narrow down the number of candidates. However, based on our data describing the DTG profile in the inflorescence of irGGPPS VIGSMaT1, and stylar DTG profiles of VIGS-MaT1 in the backgrounds of EV and irAOC, we doubt very much that analysis of irGGPPS VIGS-MaT1 styles would allow us to pinpoint these putative “key” DTGs. Even if it did, we would not be able to complement the short style phenotype to definitively test the role of these putative “key” candidates, for several reasons, which we explain below. These explanations are followed by a point-by-point response to editor and reviewer comments.

We hope that we can address the editors’ and reviewers’ concerns without conducting the additional experiments required to profile DTGs in irGGPPS VIGS-EV vs. VIGS-MaT1 styles.

Why we think that adding data on the DTG profile in VIGS-EV vs. VIGS-MaT1 irGGPPS styles will not substantially increase the explanatory value of our study:

Except for 3 of the at least 46 different DTGs (21 chemical formulas and several structural isomers), the individual compounds are not present in sufficient abundance for purification and complementation assays within a reasonable time frame; structural similarity of many DTGs further increases the difficulty (Heiling et al., 2016; Poreddy et al., 2015). Additionally, the malonylated compounds are not stable and previous attempts at the serial fractionation by UPLC of bulk extracts have failed regarding the malonylated derivatives (Heiling et al., 2016). Furthermore, depending on how much we could narrow down the candidate compounds, and starting from the 16 compounds for which we could present relative quantification in this manuscript (ignoring structural isomers), we could have up to (16! * 15!) + ((16!/2!) * 14!)) = 2.8 x 10^25^ combinations to test in complementation tests if we tested combinations of 1-2 compounds (chemical formulas) – keeping in mind that even a single candidate compound may be perceived as a ratio against one or more other DTGs. Finally, we do not have a candidate mechanism, and so if we were able to test our best guesses based on comparison of the three background genotypes (EV, irAOC, irGGPPS) with VIGS-EV or VIGS-MaT1, and these tests did not produce conclusive results, there would be no clear way to proceed.

In summary, substantial additional work is required to determine the mechanism behind the role of DTGs in the short style phenotype in *Nicotiana*, and this work would be much more rigorous if done using an approach guided by alternative mechanistic hypotheses, rather than screening as many combinations as possible of individual metabolites for their effects on stylar elongation. Currently, the number of possibilities we could rigorously test is one (Lyciumoside IV, the only abundant nonmalonylated compound which we are able to purify in quantity), which is not likely to explain the phenotype (due to its lack of malonylation).

The reviewers and the reviewing editor found that your manuscript makes an important and new contribution to plant biology. Both reviewers have a number of comments that may help improve the paper. The reviewers' comments are following below. The two reviewers have discussed their individual reviews as part of our consultation session. The major request of both reviewers is for a more detailed characterisation of the effects of NaMaT1 silencing in the backgrounds silenced for other relevant biosynthesis genes.

We agree that the characterization of the effects of VIGS-MaT1 in the backgrounds of irGGPPS and irAOC contained gaps, and we hope the revised text and new data fill these gaps and lend sufficient rigor to our study at the currently feasible level of explanatory power. The specific changes are detailed in the responses to the respective reviewer comments below.

Reviewer #1:[…] The following three issues should be addressed to strengthen the manuscript further.1) A key finding in this manuscript is that silencing NaMaT1 only results in short styles in a wild-type, but not in a NaGGPPS-silenced background. This experiment provides the main argument that the NaMaT1 effect is indeed mediated by the altered malonylation levels of DTGs, rather than some other undetected compound(s). However, it is not clear whether the style phenotype results from an excess accumulation of 'undermalonylated' DTGs, or from the altered malonylation percentage per se. It should be possible to address this by determining the malonylation percentage on the remaining DTGs in the VIGS-Mat1 irGGPPS plants. If this is still reduced to the same extent as in VIGS-Mat1 plants, it would argue that high levels of 'undermalonylated' DTGs interfere with auxin biosynthesis, rather than the malonylation level itself being critical.

We thank the reviewer for this suggestion. We analyzed the DTG profile in irGGPPS plants in existing samples, and found that the malonylation percentage for the remaining DTGs in the VIGS-*Mat1* irGGPPS plants is intermediate between EV and VIGS-*Mat1*. This new data has been added as Figure 4—figure supplement 1.

We feel it is very difficult to experimentally separate the role of “undermalonylated” DTGs from the malonylation percentage. To do so rigorously would require a tool to increase total DTG abundance without affecting malonylation percentage. The best comparison we have is the comparison of irGGPPS to EV plants (VIGS-EV); although EV has a much greater total abundance of “undermalonylated” DTGs, the two genotypes show similar malonylation percentages (new Figure 4—figure supplement 1B) have styles of similar length.

2) The reason for the sterility of the VIGS-Mat1 plants should be determined in more detail. Is it just the physical separation of stigma and anthers – in which case hand-pollination should rescue fertility – or is there also a functional difference in the styles that interferes with efficient pollen-tube growth or guidance, which might have interesting implications for plant incompatibility systems?

We thank the reviewer for this suggestion, and agree that it is important and relevant for our study to provide more information on this phenotype. We conducted this experiment and show that hand-pollination cannot rescue fertility in VIGS-MaT1 plants. We have added the data to the revised manuscript (Figure 3—figure supplement 2B, C, and D show new data; subsection “Silencing NaMaT1 affects floral development”, last paragraph).

3) YUCCA activity appears to be reduced in VIGS-NaMaT1 styles, and the authors seem to favor an effect of altered DTG malonylation levels on YUCCA enzymatic activity, rather than on YUCCA gene expression. Are YUCCA mRNA levels altered in VIGS-NaMaT1 styles?

This is an important point. We measured the transcript abundance of the style-expressed *YUCCA* by RT-qPCR, and the data showed that silencing *NaMaT1* did not affect the abundance of *YUCCA* transcripts in any background (EV, irGGPPS and irAOC; Figure 3—figure supplement 3).

Reviewer #2:[…] - The authors look at DTGs based on their malonylation amounts. It would be of interest to look at single compounds. Could the observed effect of style length be due to single compounds rather than a shift in malonylation percentage? If this is not the case data should be provided to support that only the percentage changes. In fact one could test whether the percentage matters by shifting it through the application of DTG.

We agree with reviewer’s point, and we have added the data for the individual DTGs (Figure 4—figure supplement 1).

However, combing the phenotypic and DTG profile data for VIGS in the backgrounds of irGGPPS (inflorescences), irAOC (styles) and EV (inflorescences and styles), we could not pinpoint an individual compound which clearly indicates the truncated style phenotype. For example, the dimalonylated compound Nicotianoside VII is the only individual DTG which seems to follow the same pattern as malonylation percentage in the EV and irAOC backgrounds: it is elevated in VIGS-EV irAOC vs. VIGS-EV EV styles, but reduced to similar levels in VIGS-MaT1 styles of irAOC and EV plants. However, Nicotianoside VII is reduced in irGGPPS vs. EV inflorescences, regardless of VIGS treatment (it is reduced by VIGS-MaT1 in both backgrounds). It is difficult to explain how either elevated (irAOC VIGS-EV) or reduced (EV VIGS-MaT1) levels of a single compound can cause the same phenotype. We propose that this indicates that it is a ratio, or relative abundance of multiple compounds, which is decisive. An alternative hypothesis would be that homeostasis of Nicotianoside VII (as opposed to total malonylation percentage) determines the short style phenotype. Unfortunately, Nicotianoside VII is present in low abundance relative to other DTGs and has high structural similarity to three other low-abundance DTGs: Attenoside, Nicotianoside VI, and Nicotianoside VIII (Figure 1—figure supplement 1).

Nicotianoside VII is just one candidate. As *N. attenuata* contains at least 46 different DTGs (21 chemical formulas and several structural isomers; Heiling et al., 2016), myriad DTGs changes could lead to this phenotype. Although total DTGs are as abundant as starch, Lyciumoside IV and its malonylated compounds, Nicotianoside I and Nicotianoside II, comprise more than 80% of total DTGs (Poreddy et al., 2015), meaning that other components are present in extremely low concentration. Together with the structural similarity and instability of the malonylated DTGs over time in solution, purification of some DTGs is challenging (Heiling et al., 2016). Heiling and colleagues were able to purify sufficient amounts for structural MS analysis, but bioassays require substantially more material and long-term stability under bioassay conditions.

In summary, all of these considerations make the use of most individual DTGs and mixes of DTGs in gain-of-function tests unfeasible. We found that the malonylation percentage is a uniform parameter, and all the phenotype patterns match very well with disturbances of the uniformity of malonylation percentage. Thus we think “malonylation percentage” is a good proxy for whatever the plant is specifically sensing. We agree that being able to understand exactly how plants sense “malonylation percentage” would be a high value target, it is clearly a question beyond the scope of this study, as it would likely require understanding the putative protein machinery involved in the sensing mechanism.

We thank you for these excellent suggestions and as we think that readers will also be thinking along similar lines, we have added a summary of these considerations in the revised Discussion section (fifth paragraph).

- The authors state that the phenotype is not observed with irGGPPS VIGS-Mat1 plants. No data however characterizing the chemo type of these plants is supplied. How do the DTGs levels compare to VIGS-Mat1 plants? The same goes for silencing efficiencies of Mat1. This holds for irAOC plants too. Please ensure that the reader has access to the full characterization: qRT for silencing,% of malonylation, mono-di-tri malonylation, style length, IAA and JA content, YUCCA activity for all the plants which are of importance to draw a conclusion. Please show whether YUCCA activity is inhibited in irGGPPS (+/- VIGS-Mat1) plants to support involvement of malonylated DTGs in this phenomenon. The authors claim that the mechanism is the same for irAOC plants. How do auxin levels look in those plants and is auxin biosynthesis (YUCCA) inhibited too?

We agree with the reviewer’s point about the importance of the DTG profile in irGGPPS VIGS-MaT1 plants. We reanalyzed our existing data and provided the profile of inflorescences in this version (Figure 4—figure supplement 1).

We also were able to measure the silencing efficiency in the styles of VIGS plants in background of irGGPPS, irAOC and EV, and YUCCA transcript abundance, from existing cDNA.

As mentioned above, we currently don’t have irGGPPS (+/- VIGS-MaT1) plants and style samples, and it will take 4 additional months to get flowering plants. However, we showed that irGGPPS (+/-VIGS-MaT1) didn’t affect auxin concentration (Figure 5B), we think the YUCCA activity is not crucial to address our conclusion.

The auxin levels and YUCCA activity were provided in the previous version (Figure 6F and 6G).

- Expression level of Mat1 in VIGS plants was solely investigated in leaves the place of VIGS/Agrobacterium treatment. Those are however not the organs where a phenotype was observed. Explanations about the observation of a phenotype in styles, not the place of VIGS treatment should be included. The fifth paragraph of the Discussion states the style synthesises large amounts of DTGs. Why was no qRT done on styles? AT gene expression in irGGPPS and irAOC with EV and VIGS-Mat1 should be investigated in order to draw a full picture. Please answer the question of how knock down of Mat1 affects the plants in the background of irGGPPS and irAOC more carefully. For example, don't just show malonylation percentage but show at least mono-, di- and tri-malonylation. Especially as irAOC background changes malonylation percentage to be higher.

We agree with reviewer’s points about spatial or tissue specificity, and we conducted further analyses of existing samples and added new figures (Figure 3—figure supplement 3 for gene expression; Figure 4—figure supplement 1 for DTGs pattern) to address these points.

*- The NaMaT1-3 enzymes mediate the mono- and di-malonylation of Lyciumoside IV. The mono-mannosylation however is unlikely to be the enzymes* in vivo *function as Mat1 VIGS plants produce greater amounts of the Nicotianoside I compared to EV plants. The diglycosylation, which the authors showed happens in vitro too, seems to be affected. Statements about enzyme function should be revisited in the paper as they sometimes seem rather blurry. Statements like "controls the last two malonylation steps" cannot be made as observed patterns can also be due to substrate limitations. In addition you cannot throw all the malonylated compounds together as one when talking about enzyme functions (unless you show that it really is one enzyme accepting those different substrates). Those compounds do differ greatly biochemically speaking. I would find it necessary to look at individual compounds rather than the pool of compounds categorized by mono-, di-, tri-malonylation, in order to figure out enzyme function. The specific enzyme function might help to prove the hypothesis about malonylation percentage or even show that the effect observed might be due to rather specific compounds. "found that one MaT mediates this process" is not true. Please phrase carefully. Regarding the three acyltransferases, the authors should provide details about similarity of genes/proteins mentioned in the first paragraph of the subsection “NaMaT is responsible for DTG malonylation”. Substrate specificity of NaMaT1 should be accessed especially due to observed decrease in rutin. You might not find malonylated flavonoids, but how about NaMaT1s ability to accept other acyl donors? Statements like those in the second paragraph of the Discussion do not mean anything unless tested with the protein in question. Please state how similar NaMaT1 and NbMaT1 are. Was NbMaT1 tested with DTGs?*

We agree with reviewer’s points. We re-wrote the statement about the catalytic ability of NaMaT1 in the Discussion part (second paragraph).

For the specific compounds, we performed additional data analysis and added the figure in this version (Figure 4—figure supplement 1).

The statement of “found that one MaT mediates this process” was changed to “found that one MaT contributes significantly to this process”.

We reanalyzed the malonyltransferase protein sequence and add these data in this version (Figure 2—figure supplement 1B, and subsection “NaMaT is responsible for DTG malonylation”, first paragraph).

DTGs decorated by other acyl moieties have not yet been identified in the Solanaceae (Heiling et al., 2016), and our experiment clearly showed that the truncated style phenotype is related with DTGs. Thus, whether NaMaT1 could accept other acyl donors or not is outside the scope of this work.

NbMaT1 was not tested for catalytic ability to DTGs, because we found the NaMaT2 and NaMaT3, rather than NaMaT1, are homologues of NbMaT1. Thus we think whether NbMaT1 could catalyze malonylation of DTGs is also outside of the scope of this work.

- I do not find it surprising that the malonylation percentage of DTGs are uniform. From the biosynthesis perspective if you activate the pathway all compounds accumulate (in case of co-regulated genes) and this is what we see. In contrast, if the malonylation percentage of DTGs has the ability to regulate auxin would you not expect the percentage to vary in different organs or developmental stages rather than to be uniform? From my point of view it is necessary to look at individual compounds. Statements in the third paragraph of the Discussion about the uniformity of DTG malonylation status in contrast to strong regulation of specialist metabolites is rather strange. DTGs appear to be regulated after all as they are induced after herbivory.

We agree with the reviewer that comparison of malonylation status with regulation of metabolites does not make sense, and we revised the related statement in this version (Discussion, third paragraph). We still think it is an interesting observation that the decoration of specialized metabolites seems to be maintained within uniform boundaries, and to our knowledge this phenomenon has not previously been studied.